# Investigating the environmental response to water harvesting structures: A field study in Tanzania

Jessica A. Eisma[1] and Venkatesh M. Merwade[1]

[1]Lyles School of Civil Engineering, Purdue University, West Lafayette, IN 47906, USA

*Correspondence to*: Jessica A. Eisma (jeisma@purdue.edu)

**Abstract.** Sand dams, a popular water harvesting structure employed by rural communities, capture and store water for use during the dry season in arid and semi-arid regions. Most sand dam research has been performed on the "ideal" sand dam, despite approximately fifty percent of sand dams not functioning as intended. This research involves a year-long, in-depth field study of three sand dams in Tanzania, one of which is essentially non-functioning. The study investigated a sand dam's

impact on macroinvertebrate habitat, vegetation, and streambank erosion and explored a sand dam's water loss mechanisms. Surveys of macroinvertebrate assemblage were performed each season. Vegetation surveys were performed every other month, and erosion was recorded semi-monthly. Water table monitoring wells were installed at each sand dam, and measurements were taken twice a day. The study found that sand dams are too homogeneous to provide the sustenance and refugia macroinvertebrates need at different life stages. The non-functioning sand dam has a thick layer of silt preventing infiltration

of rainwater. The functioning sand dams store a significant amount of water, but most is lost to evapotranspiration within a few months of the last rainfall. Unlike the non-functioning sand dam, the functioning sand dams have a positive impact on local vegetation and minimal impact on erosion. Sand dams can increase the water security of a community, but site characteristics and construction methods must be strongly considered to maximize the sand dam's positive impact.

## 1 Introduction

International development projects in the Global South are managed by either a national department, private company, non-governmental organization (NGO), or a group of international development agencies (Ika, 2012). Success metrics for international development projects are typically defined by the funding organization, which are most often multilateral or bilateral organizations (e.g. the World Bank, United States Agency for International Development, etc.) or individual donors (Ika, 2012). Unfortunately, project success metrics frequently tell only one side of the story, focusing on financial and technical

management rather than the social, cultural, and environmental impacts (Ika, 2012; Julian, 2016). Such a narrow definition of success omits both positive and negative unexpected consequences of international development work (Julian, 2016). Underreporting of project outcomes results in an inadequate understanding of the impact of international development work. Failure to consider whether intended long-term goals are met wastes time, money, and resources.

One example of international development projects with a questionable record of success are water harvesting structures in sub-Saharan Africa. When a specific technology's ability to improve water security is not honestly communicated along with the technology's other long-term impacts, outside organizations may embrace the technology without understanding the associated risks. Misunderstood risks lead to situations where a water harvesting technology proliferates without consideration

of project pitfalls. This has been the case with sand dams in sub-Saharan Africa. Sub-Saharan Africa is home to over 3000 sand dams, yet approximately 50 % of sand dams are essentially non-functioning (de Trincheria et al., 2018; Viducich, 2015).

Sand dams are small, reinforced concrete dams built atop impermeable streambeds in arid regions with infrequent, high-intensity rainfall (see Fig. 1). The high-intensity rainfall erodes soil from the land surface and deposits the coarser particles,

usually sand, upstream of the dam. The sand stores primarily flash flood-water, where it is naturally filtered, protected from evaporation, and helps raise the groundwater level in the surrounding area due to recharge from the increased subsurface storage (Borst and de Haas, 2006; Hut et al., 2008; Quilis et al., 2009). The extent of a sand dam's impact on the groundwater level, however, is limited by the geologic connectivity between the sand dam and the riparian zone and by the community's water use rate (Hut et al., 2008; Quinn et al., 2019). While a sand dam does filter water in a process similar to a slow sand

filter, water abstracted from sand dams via scoopholes and covered wells exceeds World Health Organization recommendations for turbidity (73 % exceedance), conductivity (24 % exceedance), and thermotolerant coliform concentration (55 % exceedance) (Quinn et al., 2018a).

Most information about sand dams comes from NGOs painting a rosy picture of the innumerable positive impacts of sand

dams. Other information on sand dams comes from studies published on one or two ideal sand dams or from sand dam models. One in-depth sand dam study examined the hydrology of a Kenyan sand dam and performed a water balance assessment of the sand dam (Borst and de Haas, 2006). Results from this study were used to develop a couple of sand dam models that explored how sand dams impact the local water table (Hoogmoed, 2007; Quilis et al., 2009). A comprehensive study of three Kenyan sand dams explored their hydrology and bare soil evaporation, while a survey of at least 50 sand dams analysed their

water quality (Quinn et al., 2018a; Quinn et al., 2018b; Quinn et al., 2019). Other studies used modelling to further explore the seepage of sand dam water through streambanks (Hut et al., 2008) and the potential of sand dams to increase water security in Ethiopia (Lasage et al., 2015). The socio-economic benefits of sand dams has also been explored (Lasage et al., 2008) along with the negative effects of sand dam siltation and/or seepage due to poor construction and or siting (Nissen-Petersen, 2006; de Trincheria et al., 2015; Viducich, 2015). Except for the Borst and de Haas (2006) study and the Quinn et al. (2018b, 2019)

studies on high-functioning sand dams, most published sand dam studies are based on survey data or modelling efforts. Published studies do not tell the whole story of sand dam impacts, and this has created a false perception of the risks involved with sand dam construction.

This study examines claims made by non-scientific bodies about sand dam impacts by investigating how diverse sand dams influence macroinvertebrate habitat, vegetation, erosion, and local water availability. Specifically, the study will investigate the following questions: (1) Are sand dams able to support macroinvertebrates? (2) What factors determine a sand dam's impact on vegetative growth? (3) How is streambank erosion affected by sand dams? (4) What are the dominant mechanisms driving water loss from the sand dams and riparian zone? Answering these questions will provide some insight into the validity of the claim that sand dams revitalize the entire ecosystem (Reversing Land Degradation and Desertification, n.d.; Sand Dams, n.d.). These questions will be explored through an in-depth field study of three sand dams in Tanzania. The sand dams are selected based on community interest in the study and diversity of dam features, such as stream width, dam effectiveness, stream valley slope, and local vegetation. This diversity of features provides a broad representation of the sand dams found throughout the region, and this study will therefore create a better understanding of how a sand dam interacts with the local environment. The study is limited to only three sand dams, because the study design relies on the active participation of local community water groups. Only three of the community water groups formed during sand dam construction remained active at the time of this study. The breadth of the study was further limited by long travel times between sites and difficulties related to equipment access.

## 2 Study Area

Tanzania is home to 55.5 million people, 70 % of whom reside in rural areas (United Republic of Tanzania National Bureau of Statistics, 2015). The climate of Tanzania varies regionally, but most of the country experiences a tropical savannah or a warm semi-arid climate (Peel et al., 2007). The northern part of the country experiences annual bimodal rainfall, with rainy seasons occurring March to May and October to December. The central and southern part of the country experiences annual unimodal rainfall, with the rainy season occurring from October to April (see Fig. 2a; Luhunga and Djolov, 2017). Tanzania is fairly flat, with the exception of the highlands on the southern border and, of course, Mount Kilimanjaro to the east of Arusha. There are at least 15 sand dams in Tanzania, three of which will serve as study sites for this research (see Fig. 2a). Most of the sand dams were funded by the Mennonite Central Committee of Tanzania (MCC) and designed by Kenya-based NGOs. Dodoma has nine sand dams; Longido, a small town near the Kimokouwa sand dam (see Fig. 2a), has four sand dams, and there are a few sand dams elsewhere in the country. The average annual rainfall for Dodoma is 601 mm, and the potential evapotranspiration is 1800 mm. The average annual temperature in Dodoma is 23.0° C. The average annual rainfall for Longido is 696 mm, and the potential evapotranspiration is 1640 mm. The average annual temperature in Longido is 20.7° C (Platts et al., 2015).

The sand dams selected for inclusion in this study all have an active community water group that was willing and able to participate in the study. The community water groups have formal ownership of the land surrounding the sand dams, and the research activities were generally limited to this land. One of the sand dams selected, Kimokouwa, was known to store very

little water outside of a few days after a rain event. The other two sand dams, Soweto and Chididimo, store water for a couple of months into the dry season. The Soweto and Chididimo sites have different site geology, and therefore provide different insights into the potential of sand dams to impact their local environment. All three sand dams vary in their construction specifications, length, and storage capacity (see Table 1). The width of the spillway is essentially equal to the width of the stream at each site.

## 2.1 Kimokouwa sand dam

The Kimokouwa sand dam (see Fig. 2b) is located approximately 11.5 km south of the Kenya border. Construction of the sand dam was completed in November 2011 with funding provided by MCC and design and construction expertise provided by the Utooni Development Organization of Kenya. The soil deposited behind the Kimokouwa sand dam is largely silty sand with thick silt layers interspersed. In the riparian zone, the soil is primarily reddish sandy clay. A hand pump was installed in the right bank, 30 m upstream of the sand dam in April 2016. MCC requested this site be included in the study, because the sand dam proved ineffective at capturing and storing water for the community's use. MCC hoped that the research could help identify the factors contributing to the sand dam's failings and inform their future work.

## 2.2 Soweto sand dam

The Soweto sand dam (see Fig. 2c) is located approximately 20 km west of Dodoma, Tanzania's capital city. Construction of the sand dam was completed in June 2011 with funding provided by MCC and design and construction expertise provided by the Sahelian Solutions Foundation of Kenya. The soil deposited behind the Soweto sand dam is moderately sorted sand, and the riparian zone is predominantly silty sand. A hand pump was installed in the left bank, 85 m upstream of the sand dam at the time of dam construction. The Soweto site is the flattest of the three sand dam sites, with an elevation change of only 14 m across the site. The streambanks are quite flat near the dam, and the community is able to grow many crops on the banks, using water from the sand dam for irrigation. At 17 m wide, the stream at Soweto is also the widest of the three sand dam sites.

## 2.3 Chididimo sand dam

The Chididimo sand dam (see Fig. 2d) is located approximately 3.2 km south of the Soweto sand dam. Construction of the sand dam was completed in June 2011 with funding provided by MCC and design and construction expertise provided by the Sahelian Solutions Foundation of Kenya. The soil deposited behind the Chididimo sand dam is moderately sorted sand. The riparian zone contains primarily silty sandy gravel. A hand pump was installed within the stream channel 150 m upstream of the sand dam at the time of dam construction. The community selected this site for the hand pump, because they were able to extract water from the sandy streambed at that location before the sand dam was constructed. The Chididimo sand dam is constructed in a fairly uniform stream valley, with relatively steep slopes covered with long grasses and large trees. The abundant vegetation is expected to reduce erosion at the site, but the steep stream valley likely means that the sand dam will have a less pronounced impact on the local water table.

### 3 Data Collection and Analysis

#### 3.1 Community water groups

Each sand dam selected for this study has an active, officially registered, community water group responsible for managing the sand dam. The community water groups were involved in the field work for this study from the first day. In addition to meeting with the researchers regularly, each group provided three to six volunteers to take twice daily and bi-weekly measurements. The volunteers were trained in proper data collection and recording procedures and were provided all materials necessary to complete the work.

#### 3.2 Macroinvertebrate survey

Macroinvertebrate surveys performed at each site were intended to serve as an indication of water quality and overall habitat health. At each sand dam, samples were extracted at two locations upstream of the dam and one location downstream of the dam. All samples were taken from the middle of the streambed. During the dry season, a 25 cm x 25 cm by 10 cm-deep hole was dug in the streambed with a small shovel, and the extracted bed material was transferred to a plastic bucket (Verdonschot et al., 2014). Holes drilled in the bucket's lid were plugged with cotton to prevent transfer of macroinvertebrates into or out of the sample (Stubbington et al., 2009). The samples were transported to the research base and rehydrated with de-chlorinated water to encourage re-emergence of desiccation-tolerant life stages (Boulton et al., 1992; Stubbington et al., 2009). For a 28 day period, the samples were checked daily for macroinvertebrates. During the rainy season and at the start of the dry season when the streambed was still fairly wet, a 25 cm x 25 cm by 10 cm-deep hole was dug in the streambed with a small shovel, and the extracted bed material was sieved through a 2 mm mesh sieve at the site. Any macroinvertebrates found would have been stored in a 10 % formaldehyde solution for later identification (Stubbington et al., 2009).

#### 3.3 Vegetation survey

Vegetation surveys were performed approximately every two months to capture the seasonal change in vegetative cover near the sand dams. The surveys were done in accordance with the line intercept method (Lutes et al., 2006). At each site, four 20 m-long transects were laid perpendicular to the stream flow and marked with wooden stakes. One transect was sampled downstream of each sand dam and three transects were sampled upstream of each dam with a 50 cm x 50 cm quadrat (Lutes et al., 2006). During each survey, the quadrat was placed consecutively along the transect, and the percent of vegetative cover was estimated visually (Lutes et al., 2006; Mallik and Richardson, 2008). At Soweto and Chididimo, quadrat 1 was placed at the stream edge and the transect extended away from the stream. Two transects were laid on the left-hand side of the stream, and two were laid on the right (see Fig. 2 b,c). At Kimokouwa, where the stream is narrow, the centre of each transect lay at the middle of the stream (see Fig. 2d).

### 3.4 Erosion study

Erosion pins were installed at each site to track the amount of streambank erosion occurring upstream and downstream of the sand dams. Welding rods 300 mm in length and 4 mm in diameter were used as erosion pins when the bank material was soft enough to insert the rods without deforming them (Lawler et al., 1999; Saynor and Erskine, 2006). The welding rods were

painted to prevent rusting (Saynor and Erskine, 2006). Stainless steel rods 300 mm in length and 6 mm in diameter were used as erosion pins elsewhere (Stott, 1997). The pins were inserted into the streambank leaving 75 mm of the pin exposed at a vertical spacing of ¼, ½, and ¾ of the bank height and at a horizontal spacing of one meter (Palmer et al., 2014). At Kimokouwa and Soweto, erosion pins were placed at two locations upstream of the dam and one location downstream of the dam. At Chididimo, pins were placed at one location upstream of the dam (see Table 2). Pins were installed at fewer locations at

Chididimo, because the stream did not have a clearly defined bank, and, where present, the streambank was often too rocky to permit insertion of the pins.

Volunteers from the community water groups took erosion measurements approximately every two weeks using a 150 mm rule depth gage. The length of pin exposed was recorded to the nearest mm for each pin. If more than 100 mm of the pin was

exposed, the pin was reset so only 75 mm was exposed. In the event that the pin was missing due to extraordinary erosion, the researchers assumed that 240 mm, or 80 % of the pin's length, of erosion occurred at the pin's location (Palmer et al., 2014). When a pin could not be found and appeared to be buried in the streambank, 300 mm deposition was assumed, and a new pin was installed with 75 mm exposed.

### 3.5 Water table monitoring

Water table monitoring wells (WTMW) were installed at each sand dam to track changes in the water table over time. A drilling team hand-augured boreholes 10 cm in diameter at 63 locations across the three sites (see Table 3). For each WTMW, the drilling team continued drilling until the team encountered hard rock or another material prohibiting the progress of the auger. A WTMW was installed only if a hole deeper than 0.5 m was achieved. A soil log was completed for each WTMW noting the soil depth, texture, colour, wetness, and cohesion for each horizon. See Fig. 2b-d for WTMW layout at the sand

dams. Fig. 3 provides a schematic of the WTMWs. The WTMWs installed were schedule 40 polyvinyl chloride pipe 32 mm in diameter. To create a well screen, four 6.35 mm holes were drilled around the circumference of the pipe every 2.5 cm, leaving the top 60 cm of the pipe undrilled (Sprecher, 2008). Geotextile filter fabric was unavailable, so the well screen was covered with women's hosiery instead (Borst and de Haas, 2006). The well caps at the top and bottom of the WTMWs were vented to prevent pressure from building up inside the pipe resulting in incorrect measurements. At the ground surface, a

mounded concrete pad was built to secure the WTMW in place and to encourage rainfall to drain away from the structure (see Fig. 3). The elevation of the top of the WTMW pipes was measured relative to the ground surface with a tape measure, accurate to the nearest cm. The ground elevation at the WTMWs was determined with a calibrated GARMIN GPSMAP 64s.

Volunteers from the community water groups took measurements of the water table every morning and evening during the rainy season after the WTMWs were installed. After the wells dried up, measurements were taken less frequently—approximately once per week. The water table measurements were taken by slowly lowering a Solinst® Model 101B Basic Water Level Meter into the WTMW until the buzzer was activated indicating water had been reached. At this point, the distance from the top of the WTMW pipe to the sensor was recorded to the nearest cm in a notebook along with the date and time of day.

At Kimokouwa, the community water group volunteers took measurements of the water depth for a few weeks after the WTMWs were installed, but water was only detected in the well closest to the sand dam up to two days following even a large rainfall event. The sand dam was clearly not storing much water. The volunteers and the researchers agreed to cease WTMW measurements at Kimokouwa so as to not waste the volunteers' time. At Soweto, the frequency and regularity of the WTMW measurements varied somewhat, with the measurements being more consistent later in the project timeline. The Chididimo community water group volunteers were very dedicated to the task of recording water table depths every morning and evening. Of the three sand dams studied, the Chididimo data provides the most complete understanding of how water storage in the sand dam changed over time.

The water table measurements were used to determine the volume of water in the sand dam and riparian zone over time. The weekly average height of subsurface water in each WTMW was calculated from the field data, accounting for the difference in soil porosity between the sand dam and the riparian zone. A value of 0.42 was used for the porosity in the sand dam; 0.40 was used for the porosity in the riparian zone (Rawls et al., 1982). Inverse distance weighting interpolation was applied to create uniformly spaced grids of average water height at a weekly time step. The weekly average water volume was then calculated by multiplying the water height grids by the grid spacing and summing across the control area. The control area is the portion of the study area enclosed by the installed WTMWs (see Fig. 2 c,d). For Chididimo, this area is 32 274 m$^2$, while it is 41 995 m$^2$ for Soweto. The weekly average control area water volume calculated from the field data is compared to a theoretical weekly average water volume, described below.

To determine the various causes of water loss from the sand dam and their relative magnitude, a theoretical water balance was calculated using data from Famine Early Warning Systems Network Land Data Assimilation System (FLDAS). FLDAS is a set of models designed to provide accurate climate estimates for the purpose of drought monitoring in data-sparse regions susceptible to food and water security issues (McNally et al., 2017). FLDAS provides daily and monthly climate data consisting of 25 different variables for Western, Eastern, and Southern Africa. In this study, FLDAS data was used as a proxy for climate data, because there is not a reliable source of climate data freely available for Dodoma, Tanzania. The theoretical water loss from the sand dam is:

$$Q_{out,dry\ season}(t) = -\alpha \times E(t) - Q_{sb}(t) - Q_{com}(t) \,, \qquad\qquad (1)$$

where $Q_{out,dry\ season}$ is the rate of water loss from the sand dam after the end of the rainy season, $E$ is total evapotranspiration modified by $\alpha$, which is 0.85, $Q_{sb}$ is baseflow-groundwater runoff, and $Q_{com}$ is the community's water use. $E$ and $Q_{sb}$ are taken directly from the FLDAS dataset (McNally et al., 2017), while $Q_{com}$ was calculated based on each community's accounting of their water use. Eq. (1) is integrated over time, $t$, and subtracted from the field data-determined volume of water in the control area at the end of the rainy season to create a theoretical volume of water curve for the sand dam area (see Fig. 9). The control area to which Eq. (1) is applied is that used for the field data water volume calculations: the portion of the study area enclosed by the installed WTMWs (see Fig. 2 c,d). The theoretical volume of water resulting from Eq. (1) has a high degree of uncertainty, because it is a simplified representation of water loss that utilizes modelled FLDAS data. However, the relative magnitude of the loss terms is likely reliable, and this is the primary focus of the conclusions that will be drawn from the model.

Total evapotranspiration, $E$, is the sum of canopy-intercepted evaporation, transpiration from vegetation canopies, and evaporation from bare soil (McNally et al., 2017). Eq. (1) is only applied during the dry season, and therefore the control volume will not lose water due to evaporation of canopy-intercepted rainfall. Including this portion of $E$ in the water balance is inappropriate. Kumar et al. (2018) found that canopy-intercepted evaporation accounts for approximately 15 % of the total evapotranspiration simulated in the Noah Land Surface Models, which are incorporated into FLDAS (McNally et al., 2017). Therefore, total evapotranspiration is reduced by 15 % in Eq. (1), resulting in an $\alpha$ of 0.85. FLDAS calculates transpiration by scaling potential evapotranspiration in proportion to solar radiation, vapor pressure deficit, air temperature, and soil moisture. Evaporation from bare soil in the FLDAS dataset is calculated by scaling potential evapotranspiration based on current soil moisture (McNally et al., 2017). Therefore, the rates of transpiration and evaporation in the FLDAS dataset will decrease as the water table retreats from the ground surface and soil moisture declines.

The community's water use was calculated using estimates provided by the community water groups, and thus has an unknown degree of uncertainty. At least one sand dam researcher has noted that unsanctioned machine pumping of water from sand dams can cause rapid drawdown of stored water (Hut et al., 2008). However, no evidence was present at either Dodoma site to indicate the community was drawing significantly more water from the sand dams than they indicated.

**4 Results and Discussion**

**4.1 Macroinvertebrates**

The various macroinvertebrate survey trials produced only one specimen—at Kimokouwa during the dry season. This failure to produce macroinvertebrates indicates that sand dams are not a suitable habitat for macroinvertebrates during any season of

the year. The absence of macroinvertebrates in the sand dams might suggest that sand dams have a negative impact on macroinvertebrate habitat, but it is also likely that sandy streambeds in semi-arid regions are simply inhospitable to macroinvertebrates. To make this distinction, further studies are needed to compare macroinvertebrate assemblages in undammed sandy streambeds with those in sand dams. Of all substrates studied, Duan et al. (2008) identified sandy substrate

to have the lowest taxa richness and to be the least suitable for macroinvertebrates and benthic fauna, causing sandy substrates to be fairly homogeneous. Sandy substrate also has small interstice dimensions that provide only very small living spaces for macroinvertebrates (Duan et al., 2008). Homogeneous bed material suggests that there are few, or no, structures available for macroinvertebrates to use as refugia during high streamflow (Taniguchi and Tokeshi, 2004) and few niches for different species to utilize during various stages of their life cycle (Salant et al., 2012). Furthermore, macroinvertebrates feed on bacteria, algae,

and other organic matter, which may be scarce in sandy substrate (Taniguchi and Tokeshi, 2004).

In the case of the sand dams studied, there were very few plants, cobbles, or larger rocks present in the stream channel. The sand within the sand dam, with the exception of Kimokouwa, was largely a mixture of fine- and coarse-grained sand, as determined by a visual and tactile assessment of the material. This environment precluded macroinvertebrates from inhabiting

the sand dam. Macroinvertebrates are often used as an indicator of water quality, but the lack of macroinvertebrates in the sand dams here should not be assumed to signify the water was of low quality. The aforementioned compounding factors likely largely explain the absence of macroinvertebrates in the sand dams.

### 4.2 Vegetation

The vegetative cover at the three sand dams differed greatly throughout the study (see Fig. 4). Kimokouwa had the lowest level

of vegetative cover overall and did not exhibit much increase in vegetative cover during the rainy season. Soweto, the flattest site, showed the greatest improvement in vegetative cover between the dry season and the rainy season. Each Soweto transect exhibits a significant increase in vegetation. Interestingly, Chididimo only had significantly more vegetation at the two transects farthest upstream from the sand dam (VT3 and VT4). The slope of the Chididimo stream valley became gentler farther upstream of the dam (see Fig. 2d), which created favourable conditions for increased vegetation during the rainy season.

Of the three sand dams, Chididimo has the highest level of vegetation during the dry season, and therefore had the least opportunity for significant increases in vegetation during the rainy season.

That Soweto, the flattest site, and the two transects located in the flattest part of Chididimo display significant increases in vegetation between the dry and rainy seasons suggest that the average percent vegetative cover at a sand dam is correlated to

the land slope near the sand dam. The Pearson Correlation Coefficient, $\rho$, corroborates this observation. The change in vegetative cover between the dry and rainy seasons is negatively correlated ($\rho = -0.73$) to increasing land slope at the two functioning sand dams, Soweto and Chididimo, indicating that as the land slope increases, the improvement in vegetative cover

decreases. The same correlation is not observed at the non-functioning Kimokouwa sand dam ($\rho = 0.04$), which is expected because the sand dam is not contributing to a locally raised water table.

As the elevation above the streambed increases, the percent vegetative cover generally decreases during both the rainy season and the dry season (see Fig. 5a). The trend of decreasing vegetative cover with increasing elevation above the streambed is more consistent during the dry season but is also evident during the rainy season. Two conditions may combine to create the trend seen in Fig. 5a. First, at low elevations above the streambed, groundwater seepage through the streambanks creates a raised water table that is close to the land surface (see Fig. 6). The raised water table has a positive impact on the soil moisture of the unsaturated soil layer, and this additional moisture supports vegetation growth. Second, a lower elevation above the streambed implies a gentler land slope. Gentle slopes give rainwater more time to infiltrate into the soil, because storm surface runoff travels slower over a gentle slope. Increased infiltration results in increased soil moisture and increased recharge of the water table. As Fig. 5a indicates, there is low vegetative cover right at the stream edge (lowest elevation), which signifies streamflow frequently rising above this point and inhibiting vegetation growth.

That the dry season shows a consistent relationship between elevation above the streambed and vegetative cover indicates that the vegetation at Soweto and Chididimo has at least some level of groundwater dependence (see Fig. 5a). The dependence of vegetation on groundwater in arid and semi-arid regions has been well-documented (Elmore et al., 2008; Mata-González et al., 2012; Naumburg et al., 2005; Seeyan et al., 2014; Stromberg et al., 1996; Wang et al., 2011). In arid and semi-arid regions where rainfall is minimal, vegetation often relies on groundwater to supply the additional water needed for plant growth and transpiration (Naumburg et al., 2005). In semi-arid Dodoma, local communities use their knowledge of the relationship between vegetation and groundwater to inform their decisions on where to dig shallow wells (Shemsanga et al., 2018). Therefore, it is reasonable to expect that the vegetative cover at the Soweto and Chididimo sand dams is improved, in part, by a locally raised water table near the ground surface.

The upstream and downstream vegetative cover trends differ at the three sand dams. At each sand dam, there is more vegetation upstream of the sand dam than downstream (see Fig. 5b). However, this difference is most significant at Soweto, where the change in elevation across the site is small relative to the Kimokouwa and Chididimo sand dams. Of the three sand dams studied, only the sand dam located in a flat area exhibited a large increase in vegetation upstream of dam, indicating that a sand dam's impact on vegetation may be limited by the slope of the surrounding land. Nevertheless, additional studies are needed to verify this relationship. Also, the rate of vegetative cover at the two functioning sand dams, Chididimo and Soweto, is high compared to the non-functioning Kimokouwa sand dam. This may be due solely to the impact of the sand dams, but it is equally likely that the steeper slopes and finer soils at Kimokouwa impact its vegetative cover. A functioning sand dam has the potential to support more vegetation, because of the additional stored water that is available to vegetation for use in transpiration. Land slope and soil, however, must also be considered.

### 4.3 Streambank erosion

The temporal changes in the bank soil varied somewhat across the three sand dams (see Fig. 7). Kimokouwa and Soweto exhibited little change in bank volume at the upstream locations, and Chididimo experienced a high rate of soil deposition. Interestingly, bank erosion increased at Kimokouwa during the rainy season, while bank deposition increased at Chididimo during the rainy season. The differences in bank morphology and floodplain vegetation between the two sites impact their respective erosion/deposition dynamics. At the downstream location, Soweto did not exhibit much change in bank soil, while the Kimokouwa site showed severe erosion, particularly during the long rains season (mid-February to April). The Kimokouwa downstream bank lost a total of nearly 300 mm of soil throughout the course of the study due to mass bank failure. The heavy rainfall during the long rains season led to a pre-wetted bank with heightened pore water pressures. Eventually, the pore water pressures exceeded the structural integrity of the bank, and the bank material fell into the stream channel in large volumes (Hooke, 1979; Lawler et al., 1999). The downstream Kimokouwa bank experienced multiple mass failures throughout the long rains season (see Fig. 7).

The spatial changes in bank soil also vary between the three sand dam sites (see Fig. 8). The Kimokouwa streambanks exhibit a consistently high rate of erosion across the entire bank height. The high rate of erosion is likely due to the relatively steep and/or vertical banks and minimal vegetative cover. At Chididimo, the streambank generally experiences deposition across the bank height, but does experience lower rates of deposition at the foot of the bank with some periods of erosion occurring. This is clear from the high standard error for ¼ bank height at Chididimo. Erosion at the foot of the Chididimo streambank is caused by high streamflow during the rainy seasons. At Soweto, soil eroded from the top of the bank is deposited at the middle and foot of the bank. However, the extremely long standard error bars for Soweto erosion measurements at all bank heights challenge the validity of the Soweto erosion data. The community volunteers at Soweto may have erroneously recorded the erosion measurements, despite repeated training and practice sessions with the primary field researcher. The Soweto erosion data should be considered sceptically. However, based on Soweto erosion data taken solely by the primary field researcher, the overall trend of little erosion and deposition occurring at Soweto can be confirmed.

The Kimokouwa sand dam was constructed in an unstable reach. The stream channel is actively migrating, which causes the stream to flow into the left wing wall of the sand dam, rather than flow over the spillway. A strong eddy develops, eroding the soil directly behind the dam. This erosion threatens the stability of the dam, because the dam's design depends on the weight of the soil to help hold the dam in place. The migration of the stream channel likely contributes to the mass erosion of the bank downstream of the sand dam (see Fig. 7).

## 4.4 Water storage and loss

The sand dam at Kimokouwa has a 1.2 m thick silt layer beginning at a depth of 0.5 m that acts as a capillary barrier, inhibiting the infiltration and, therefore, storage of water in the sand dam. Kimokouwa sand dam's water storage is also likely limited by the poor connectivity between the silty sand in the channel and the reddish clay that dominates the riparian zone. Groundwater
is unable to travel freely between the sand dam and the riparian zone, as evidenced by the absence of water in all but one WTMW. As a result of limited storage, the community is unable to use the sand dam as a source of domestic water. Silt layers formed at the Kimokouwa sand dam, because the dam was improperly constructed for the type of topsoil present in the area (Nissen-Petersen, 2006; de Trincheria et al., 2015). While the literature on this topic is not well-developed, the soil composition of the streambed before a sand dam is constructed can likely be helpful in determining the distribution of grain sizes a sand
dam is expected to capture. This information, coupled with knowledge of the sediment load typically carried by the stream, can inform the need to construct a sand dam's spillway in stages to prevent siltation. Siltation of a sand dam occurs during rainfall events prior to the sand dam's maturation, or before the sand reservoir has naturally reached the height of the spillway (see Fig. 1). Sand dams in areas with silty sand should be constructed in thirty cm stages to ensure that the portion of the water column with suspended silt flows over the spillway instead of settling behind an immature sand dam (Nissen-Petersen, 2006).

A functioning sand dam typically fills with water after one high intensity rainfall event and remains essentially full throughout the rainy season (Ertsen and Hut, 2009). The stored water seeps into the banks, raising the water table in the riparian zone. The last rainfall of the season at Chididimo and Soweto occurred around early to mid-April, approximately the fourteenth or fifteenth week of the year. Fig. 9a shows that within just ten weeks of the last rainfall, the Chididimo sand dam had dried
significantly, leaving very little abstractable water available to the community. The Soweto sand dam has a much greater storage capacity and retains abstractable water for approximately fifteen weeks after the last rainfall (Fig. 9b). Soweto's greater storage is due to the wider and deeper sand reservoir. The sand dams at Chididimo and Soweto only store water for community use during the first few months of the dry season.

In Chididimo, there are three sources of water: the sand dam and two boreholes drilled by an international non-profit organization. When there is water in the sand dam, the community draws all water for agricultural use from the dam and about half of the domestic water from the dam, totalling 15 000 litres of water per week (Chijendelele na Mlimo Group, personal communication, May 30, 2017). However, their total water use accounts for only about 10 % of the water stored by the sand dam at the end of the rainy season. Unsurprisingly, most of the water in the sand dam is lost to evapotranspiration (ET). With
only 2 % of the total water lost during the dry season attributed to baseflow-groundwater runoff, ET was responsible for the remaining 88 % of the water lost from the Chididimo sand dam according to FLDAS data and Eq. (1) (see Fig. 9a). Eq. (1) predicted the sand dam would lose its stored water by week 23. The measured data indicates the sand dam retained water until

week 27. Furthermore, Fig. 9a shows that the sand dam experienced a loss reduction of 400 000 litres during the first twelve weeks of the dry season. This suggests the dam effectively reduced ET by 19 % compared to the ET simulated by FLDAS.

The nearly constant decrease in water volume, or total loss rate ($L_T$), after the end of the rainy season at the Chididimo sand dam indicates that most water is lost due to ET (see Fig. 9a). Chididimo's relatively shallow sand reservoir results in all ET occurring at a shallow ET rate. When ET occurs from a sub-surface water table, the rate of ET is lower than would be expected if the water table was at the ground surface (Hellwig, 1973). The rate of sub-surface ET decreases as the water table retreats farther underground, and the rate of decrease is dependent upon depth and grain size (Hellwig, 1973; Quinn et al., 2018b). Seepage could contribute to the total loss rate at Chididimo. While there was no evidence of seepage under the dam wall, downward seepage through the streambed could impact the total loss rate at Chididimo.

In Soweto, the sand dam is the only nearby source of water. When the sand dam is dry, community members must travel seven km to draw water from a well in a nearby village. When able, the community draws approximately 39 000 litres of water from the sand dam per week for both agricultural and domestic use (Vumilia Group, personal communication, June 1, 2017). Their total water use accounts for only about 10 % of the water stored by the sand dam at the end of the rainy season. With only 1 % of the total water lost attributed to baseflow-groundwater runoff and 16 % of loss unaccounted by FLDAS, ET was responsible for 65 % of the water lost from the Soweto sand dam according to Eq. (1) (see Fig. 9b). The unaccounted water loss could be due to seepage under the sand dam wall, through the streambanks, or streambed combined with a lower rate of ET than simulated by FLDAS.

The Soweto dam exhibits three distinct phases of water loss: shallow ET, deep ET, and minimal ET (see Fig. 9b; Quinn et al., 2018b). The minimal ET phase occurs during the period in which the community water group indicated they were no longer able to abstract water from the sand dam. At this point, the water table has retreated too far underground for the community to draw water and, at this depth, the rate of ET is likely negligible (Hellwig, 1973; Quinn et al., 2018b). Therefore, most of the water lost during the minimal ET phase is lost due to seepage under the dam wall and/or through the streambed. Unlike Chididimo, the Soweto sand dam does exhibit evidence of seepage—the community members collect water from scoopholes they dig just downstream of the dam. The seepage loss at Soweto occurs at a rate of approximately 0.2 mm day$^{-1}$ and accounts for 24 % of the water stored by the sand dam at the end of the rainy season. The seepage rate is assumed to remain essentially constant throughout the shallow, deep, and minimal ET phases (see Fig. 9b). Having accounted for seepage, Fig. 9b shows that the sand dam experienced a loss reduction of 600 000 litres during the first 29 weeks of the dry season. This suggests the dam effectively reduced ET by 11 % compared to the ET simulated by FLDAS.

Fig. 9 shows that the sand dams lost water at a slower rate than predicted by Eq. (1) during the dry season. The FLDAS dataset calculates evaporation from bare soil based on simulated soil moisture content (McNally et al., 2017). However, the dataset

does not account for the depth at which the sand dam water is stored or for unique features, such as wind speed, topography, vegetation, and shading, that impact ET rates (Hellwig, 1973; Quinn et al., 2018b). The lines fit to the field data after the end of the rainy season indicate that the Chididimo and Soweto sand dams are losing water primarily via ET at a nearly constant rate of 0.7 mm day$^{-1}$ and 1.6 mm day$^{-1}$, respectively. Due to ET and other major losses, the sand dams can no longer provide

water to the community after the months of July or August in most years.

The Soweto sand dam is losing water during the shallow ET phase at more than twice the rate of the Chididimo sand dam. The combination of stream width and vegetative cover contribute to Soweto's higher rate of water loss. The width of the Soweto sand dam is nearly twice that of Chididimo, providing a greater surface area of sand from which evaporation occurs (see Table

1). For an equivalent water content, sub-surface evaporation rates from sand in the sand dam are higher than from the loamy soils in the riparian zone due to the differences in soil suction (Wilson et al., 1997). Also, different types of vegetation transpire water at different rates (Lautz, 2008). The banks of Chididimo are generally covered with natural vegetation, whereas the Soweto community intensively cultivates the banks. Natural vegetation in a semi-arid climate requires less water than cultivated crops, therefore the rate at which the Soweto vegetation transpires water contributes to Soweto's rapid water loss.

One of the most common reasons given for building sand dams is that they provide water to communities throughout the dry season. At Chididimo and Soweto, this is simply not the case. Chididimo and Soweto experience approximately 100 mm lower annual rainfall in one, four-month rainy season and higher rates of ET than a typical sand dam in Kenya experiences during two rainy seasons (NASA/GSFC/HSL, 2016). Therefore, the Dodoma sand dams have lower potential for storing water than

their Kenyan counterparts. Sand dams are intended to protect the stored water from evaporation and they do to some extent, but the ground surface is inadequate protection against the high temperatures and dry air at the Chididimo and Soweto sand dams.

## 5 General Discussion and Considerations

The impact of a sand dam depends not only on its dimensions and construction but also on features of the surrounding land

and the management of the dam's water resource by the local community. The field study revealed that a non-functioning sand dam might significantly influence streambank erosion but has little impact on the local water storage and vegetation. The functioning sand dams, however, had little impact on streambank erosion, significant impact on local water storage and in reducing ET losses, and varied impact on vegetation. Regardless of a sand dam's functionality, none of the sand dams in the study were a suitable habitat for macroinvertebrates. The absence of macroinvertebrates in sand dams may limit the value of

ecosystem services, such as nutrient cycling, decomposition of organic matter, or primary productivity. The lack of these services, however, may be outweighed by the increased water security.

The two functioning sand dams support more vegetation than the non-functioning sand dam. The increase in vegetation caused by the sand dam's additional stored water is much more apparent when the surrounding land is relatively flat (i.e. at Soweto). A locally raised water table in a flat area results in soil water that is closer to the land surface than if the sand dam were surrounded by steep slopes, like at Chididimo (see Fig. 6). The increased soil water near the land surface is available to support vegetative growth and transpiration, leading to higher vegetative cover. Holding all other variables constant, building a sand dam in a flat area would likely maximize the positive impact of the sand dam on local vegetation. However, this needs to be further explored to see if the trend holds when more sand dams are examined.

The two functioning sand dams have stable streambanks compared to the non-functioning Kimokouwa sand dam. The streambanks at Kimokouwa exhibit severe erosion, particularly at the site downstream of the sand dam. The Kimokouwa sand dam was constructed between two sharp bends in the stream, and the flow of water over the sand dam adds energy to the water in the stream. With this added energy, the water erodes more of the streambanks and likely contributes to the migrating of the Kimokouwa stream channel. Severe streambank erosion and/or stream migration can lead directly to sand dam failure by weakening the soil supporting the structure. When this happens, the dam may break or be washed downstream. Sand dams should probably be built in stable, straight reaches to minimize the chance that the construction of a sand dam will negatively impact the course of the stream.

While the non-functioning Kimokouwa sand dam does not increase the availability of water in the local community, the functioning sand dams provide a local water resource for at least the first few months of the dry season. However, the two functioning Dodoma sand dams do not store water throughout the entire dry season, as is an often-stated benefit of sand dams. Dodoma lies in the unimodal rainfall region of Tanzania, whereas Kenyan sand dams experience bimodal rainfall. With only one period of rainfall refilling the sand dams every year, the Dodoma sand dams are unable to supply water throughout the eight-month dry season. In addition, the Dodoma sand dams receive approximately 100 fewer mm of rainfall every year and average higher rates of ET than the annual rainfall and average ET at the Kenyan sand dams. Less rainfall limited to one rainy season of the year and higher ET result in sand dams that function at a lower level than those in Kenya.

A frequently cited benefit of sand dams is that they protect the stored water from ET. While the Chididimo and Soweto sand dams helped slow the rate of ET by 19 % and 11 %, respectively, ET is still the greatest loss factor for the stored water. Chididimo has a shallower sand dam and lost 88 % of its stored water to ET, while Soweto is deeper and lost 65 % of its stored water to ET. The deeper Soweto sand dam lost less water to ET than the shallower Chididimo sand dam, because the rate at which sub-surface water can be evaporated depends, in part, on the depth of the water below the ground surface (Hellwig, 1973; Quinn et al., 2018bf). To help reduce the amount of water lost from sand dams due to ET, sand dams should likely be constructed in locations where a deep sand reservoir can develop. At least one other study, de Trincheria et al. (2015), also recognised the impact of shallow sand reservoirs on water lost to ET.

## 6 Future Work

Future analysis of the collected dataset will focus on exploring the spatial variability in the local geology and its interactions with groundwater in the vicinity of the sand dam. Groundwater dynamics will be investigated in conjunction with the variability of evapotranspiration in and around the sand dams. In addition, the change in vegetative cover relative to groundwater depth will be studied using both the field measurements detailed here and Normalized Difference Vegetation Index. Future sand dam research should also investigate water quality in the sand dams in the context of increasing salinity as evidence for or against high rates of evapotranspiration.

### Data Availability

The raw sand dam data has been published in the Purdue University Research Repository and can be accessed at: https://www.doi.org/10.4231/GYSC-1X41 (Eisma and Merwade, 2019). FLDAS data can be downloaded at: https://disc.gsfc.nasa.gov/datasets?Keywords=FLDAS.

### Author contribution

Jessica Eisma collected and analysed data during the field studies and prepared the manuscript. Venkatesh Merwade supervised the data analysis and contributed to the manuscript.

### Competing interests

The authors declare that they have no conflict of interest.

### Acknowledgments

We would like to thank the Mennonite Central Committee of Tanzania, especially Al Wright, for the invaluable introduction to Tanzania's sand dams and their community water groups. We would also like to thank our translators, Christina Sumayani, George John Filex, and Benedict Mwiliko, for their tireless assistance in all matters. Further gratitude goes to the Kimokouwa, Soweto, and Chididimo community water groups, without whom this research would not have been possible. Additional appreciation goes to Nelson Mandela African Institute for Science and Technology, especially Prof. Karoli Njau, for providing a home base and instrumental guidance. This work was supported by the NSF grant DGE-1333468 (Graduate Research Fellowships Program) and the USAID grant A1134 to Purdue University. Further support for this work came from a Fulbright U.S. Student Award to Jessica Eisma.

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

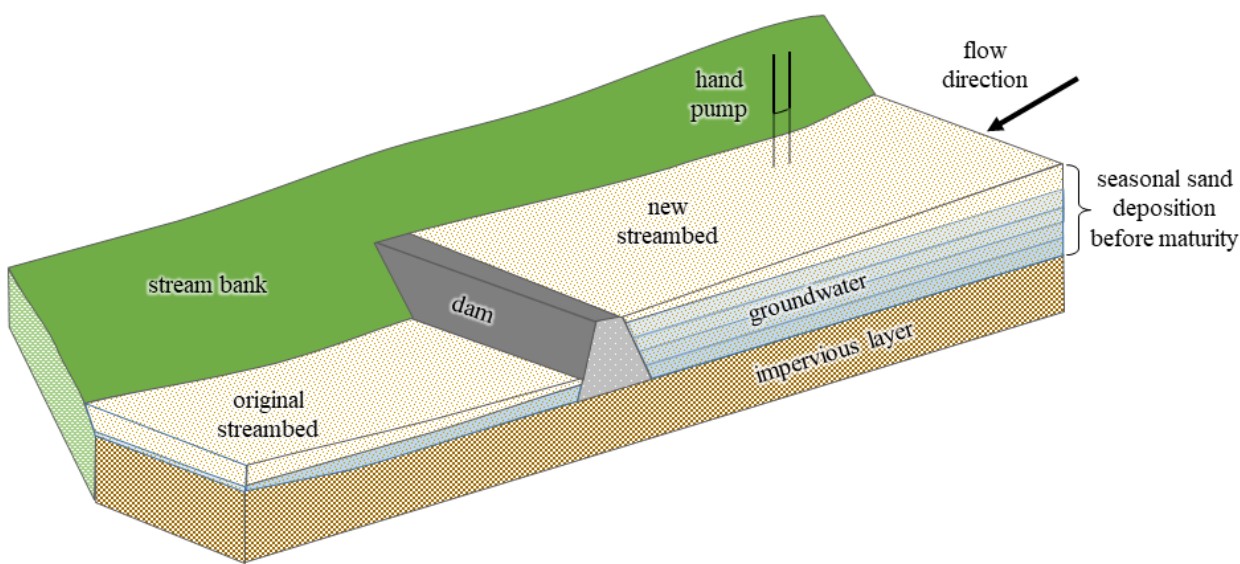

**Figure 1: Schematic of a sand dam showing seasonal sand deposition before the dam reaches maturity (adapted from Borst and de Haas, 2006).**

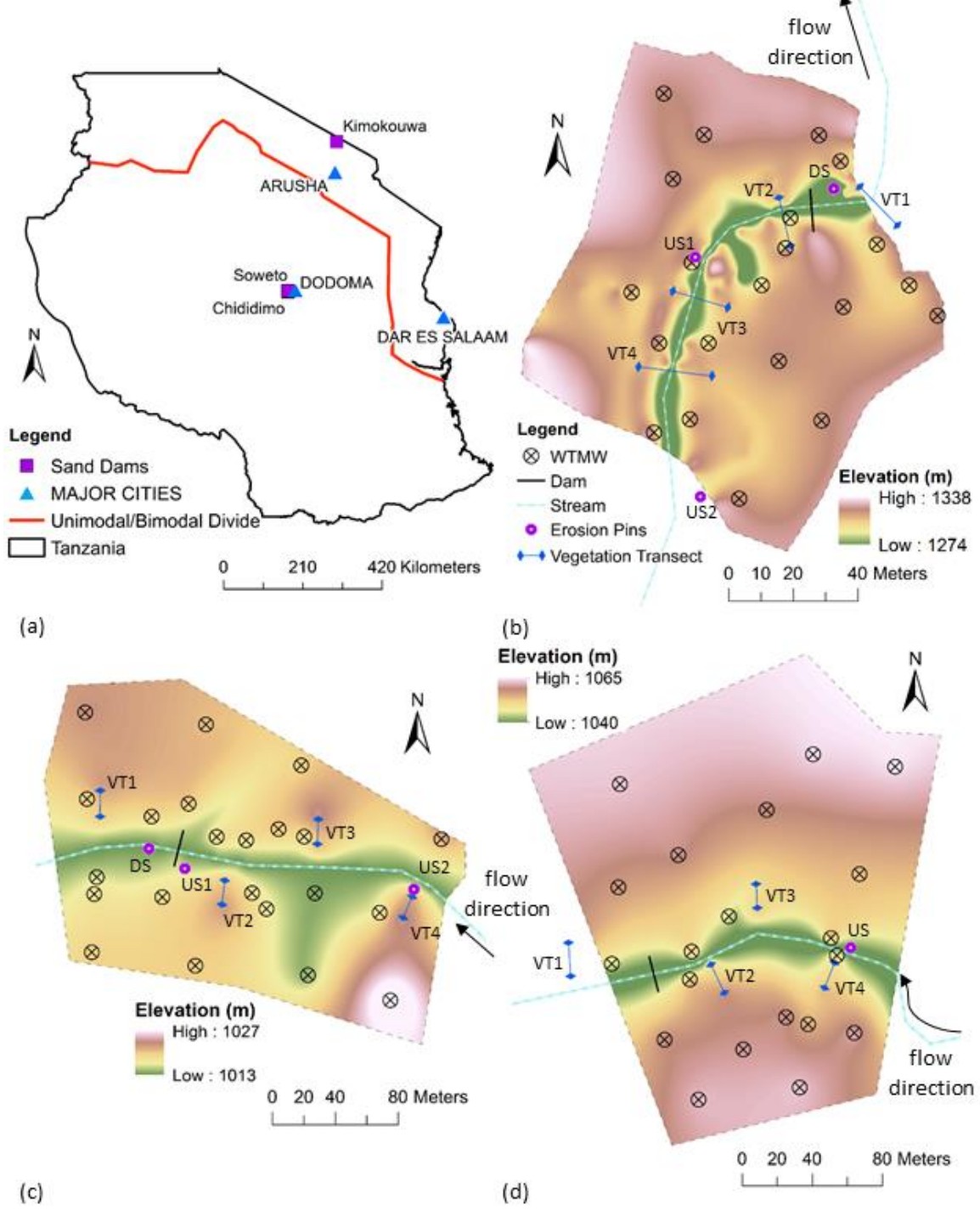

**Figure 2: (a) Study Area. The bimodal rainfall region is north of the red line; the unimodal rainfall region is south of the red line (Luhunga and Djolov, 2017); (b) Kimokouwa study area; (c) Soweto study area; (d) Chididimo study area. Elevations are interpolated from GPS points taken during study. The elevation map includes only the area controlled by the community water groups.**

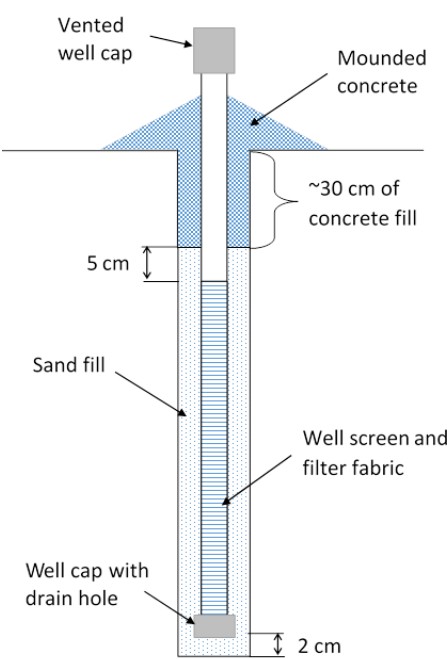

**Figure 3: Schematic of the water table monitoring wells installed (adapted from Sprecher, 2008).**

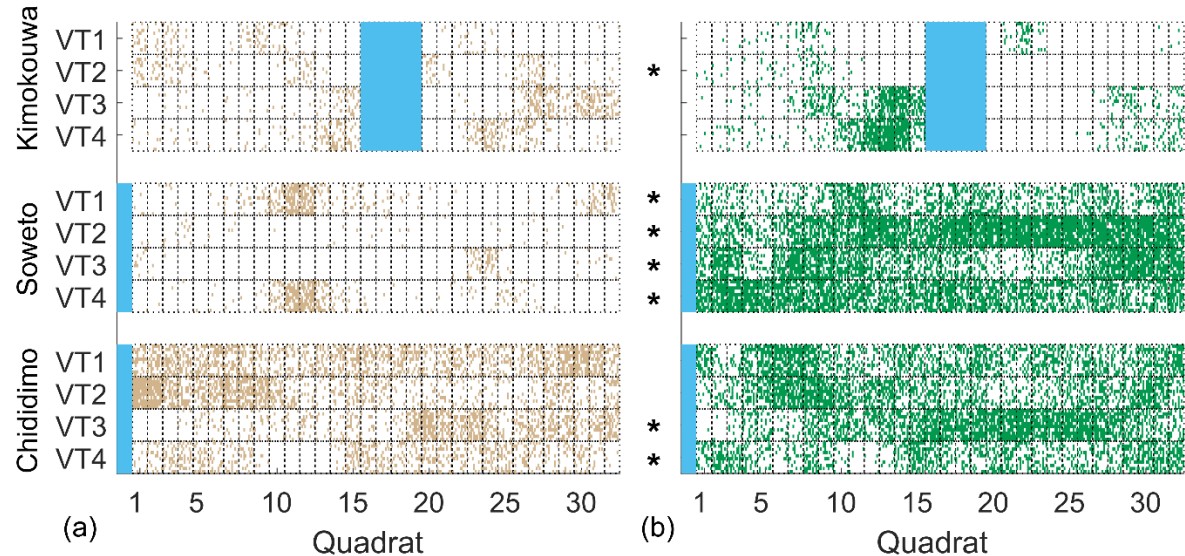

5    **Figure 4: Representation of percent vegetative cover for each transect at each sand dam during the (a) dry season and (b) rainy season. The stars indicate a significant difference (p<0.05) between the wet and dry season vegetative cover for that transect. The solid colour indicates the location of the stream relative to the transect. VT1 is downstream of the dam; VT2–4 are upstream.**

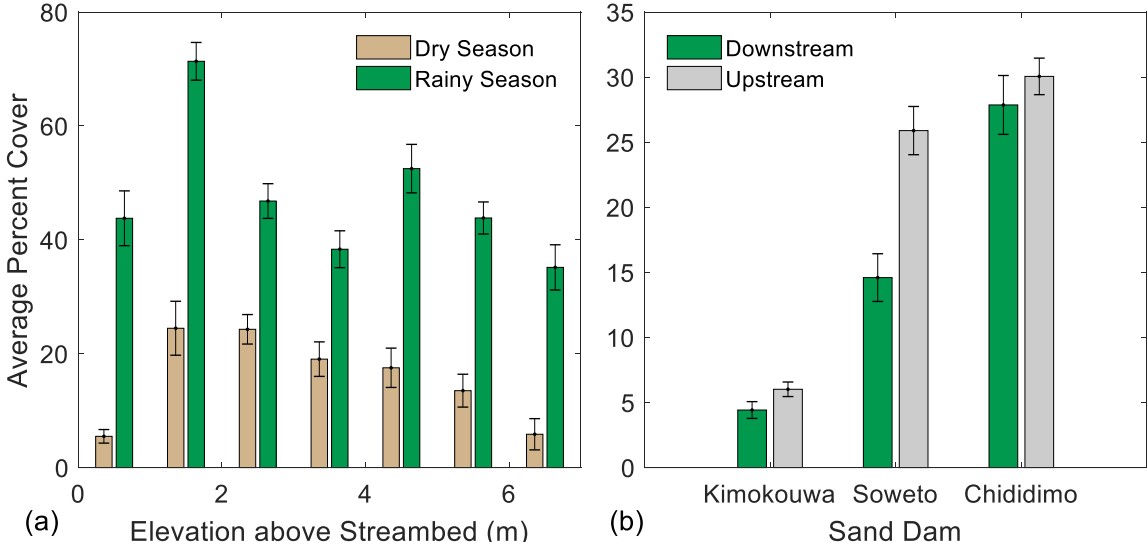

**Figure 5: (a) Average percent cover at different elevations above the streambed for the dry and rainy seasons at the Soweto and Chididimo dams. Kimokouwa sand dam was excluded, because the sand dam is not functioning. Standard error bars are shown; (b) Average upstream and downstream vegetative cover at the three sand dams. Standard error bars are shown.**

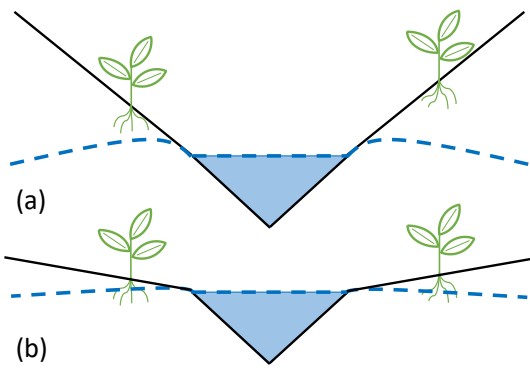

**Figure 6: The roots of plants growing on a (a) steep slope will be farther from the locally raised water table created by a sand dam, and therefore have less access to soil water, than vegetation growing on a (b) gentle slope.**

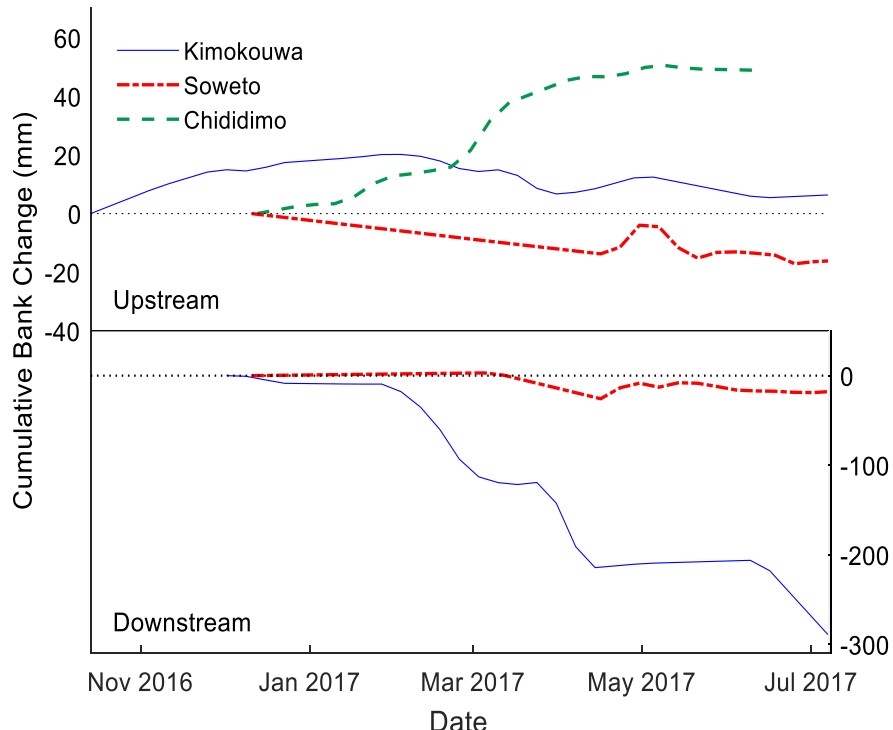

**Figure 7: Cumulative bank change over the duration of measurement at each sand dam for the upstream and downstream pinned banks. A positive value signifies deposition, a negative value indicates erosion.**

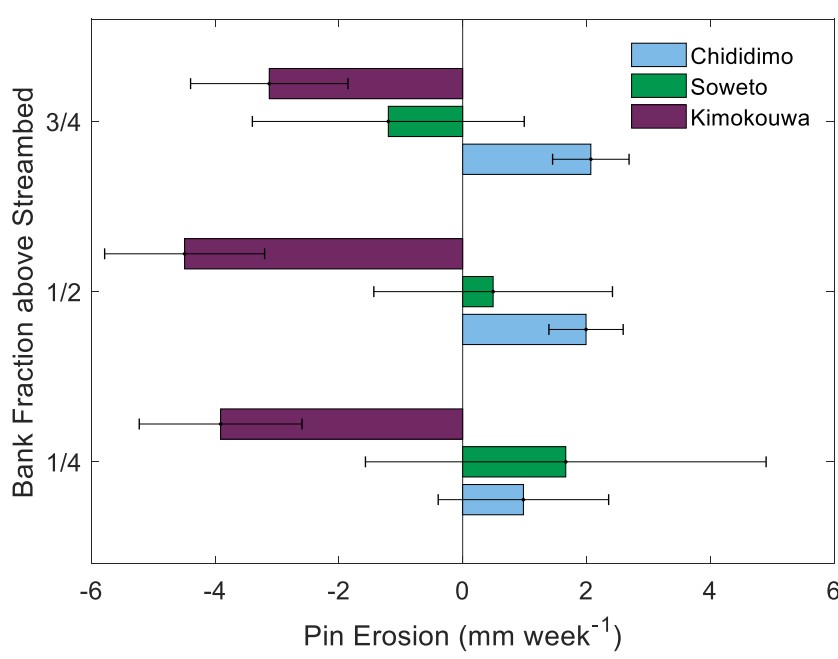

**Figure 8: Average weekly change in the bank soil at ¼, ½, and ¾ bank height. Positive values represent deposition; negative values represent erosion. Standard error bars are shown.**

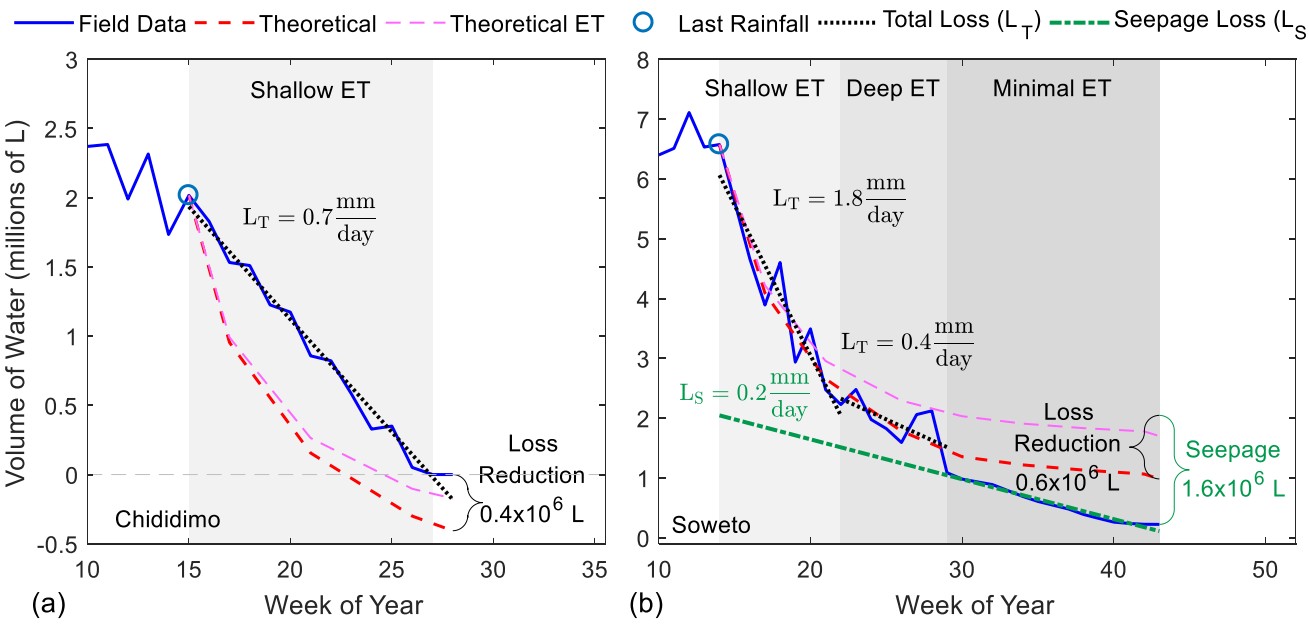

(a)

(b)

Figure 9: Volume of water in the area enclosed by the WTMWs of the (a) Chididimo and (b) Soweto sand dams. The field data line shows the volume of water in the study area during the specified week. The theoretical line, initiated at the end of the rainy season, shows the theoretical volume of water in the study area, calculated by integrating Eq. (1) and subtracting from the field-determined volume of water at the end of the rainy season. The theoretical line accounts for losses due to evapotranspiration, baseflow-groundwater runoff, and community use. The theoretical ET line shows the portion of total theoretical loss attributed to ET in the FLDAS dataset.

**Table 1:** Physical parameters of the three sand dams

| Sand Dam | Total width (m) | Total length (m) | Spillway (m) | Estimated storage volume* (m³) | Wing walls (m) | Spillway height (m) |
|---|---|---|---|---|---|---|
| Kimokouwa | 28.78 | 150 | 8.74 | 1310 | 20.04 | 2.06 |
| Soweto | 23.96 | 350 | 16.95 | 5930 | 7.01 | 1.27 |
| Chididimo | 22.71 | 300 | 9.60 | 2880 | 13.11 | 1.30 |

*Note: Storage volume estimated using an average sand dam depth of 2.5 m and porosity of 0.40. The spillway is approximately equal to the width of the stream channel.

**Table 2:** Site characteristics of each erosion pin section

| Sand Dam | Site Name | Bank | Length of bank pinned (m) | Bank height (m) | Bank morphology | Floodplain vegetation | Bank material | No. of pins | Pin spacing (m) V | H | Date installed |
|---|---|---|---|---|---|---|---|---|---|---|---|
| Kimokouwa | US1 | left | 5 | 3.7 | gently sloping | sparse trees and long grasses | silty sand | 13 | 0.9 | 1 | 14/10/16 |
| | US2 | right | 6 | 5.2 | composite | | silty sand and gravel | 18 | 1.3 | 1 | 8/12/16 |
| | DS | left | 5 | 1.0 | concave | | clayey sand | 15 | 0.2 | 1 | 24/11/16 |
| Soweto | US1 | left | 4 | 2.4 | vertical | large bushes and trees | silty sand | 12 | 0.6 | 1 | 11/12/16 |
| | US2 | left | 4 | 2.7 | steeply sloping | long grasses, small bushes and trees | clayey sand and gravel | 12 | 0.7 | 1 | 11/12/16 |
| | DS | left | 6 | 3.7 | composite | long grasses and bushes | sand | 18 | 0.9 | 1 | 11/12/16 |
| Chididimo | US | right | 9 | 3.7 | composite | long grasses and bushes | sand | 26 | 0.9 | 1 | 13/12/16 |

*Note:* US is upstream; DS is downstream; V is vertical (pin spacing); H is horizontal.

**Table 3:** Water table monitoring well installations at the three sand dams

| Sand Dam | Number installed | Range of depths (m) | Average (m) | Standard deviation (m) |
|---|---|---|---|---|
| Kimokouwa | 21 | 0.6–2.6 | 1.4 | 0.6 |
| Soweto | 22 | 0.5–3.7 | 1.5 | 0.8 |
| Chididimo | 20 | 0.5–1.9 | 1.0 | 0.4 |

