# Peer review of "Investigating the environmental response to water harvesting structures: A field study in Tanzania"

_Hydrology and Earth System Sciences, 2019_

## Short Comment (SC1) · 6 May 2019

Dear authors,

I am posting one comment about your paper, in particular on line 6, page 2 "yet approximately 50% of sand dams are essentially non-functioning (Viducich, 2015)"

Let me premise that I worked on sand dams with a small research, and that, however, I really like the critical view of your paper. The figure of 50% of sand dams not working shocked me a little bit and I revised the source, that is quite a good MSc thesis, https://ir.library.oregonstate.edu/concern/graduate_thesis_or_dissertations/1z40kx51c,

but however represents a "grey source".

Moreover, the thesis analyses only 11 Sand dams sites in a region of Kenya.

The statement reported in your thesis is found in the conclusions Page 86 of the thesis "Sand dams represent a viable rainwater harvesting and storage solution for many rural communities, but they are not universally appropriate. Performance varies widely, with reported failure rates in the range of 50%." But it seems not referenced, not either justified by the results (maybe is the 50% of the study sites).

I think this statement, which is quite heavy considering that is a major claim supporting your paper, can be revised, or maybe cited referring to other literature.

Hope this comment can be of your interest,

Kindest Regards Giulio Castelli

---

## Author Comment (AC1) · 6 May 2019

Dear Giulio Castelli,

Thank you for your comment on the manuscript.

In response, we have returned to the literature to provide stronger justification for the statement that 50% of sand dams are essentially non-functioning. de Trincheria et al. (2018) support the statement made by Viducich (2015) and further state that the estimate of 50% non-functioning sand dams may be conservative. de Trincheria et al. (2018) develop this conclusion based on their study of 30 sand dams in southeastern

Kenya and knowledge of hydrology, geology, and construction practices. We would also like to note that the estimate seems reasonable based on our personal experience with sand dams in Tanzania and interactions with the local communities. To support the claim, de Trincheria et al. (2018) will be added as a citation to the relevant sentence, and the following will be added to the reference list:

de Trincheria, J., Filho Leal, W., Otterpohl, R.: Towards a universal optimization of the performance of sand storage dams in arid and semi-arid areas by systematically minimizing vulnerability to siltation: A case study in Makueni, Kenya, Int. J. Sediment Res., 33, 221-233, doi:10.1016/j.ijsrc.2018.05.002, 2018.

Considering your comment, we would also like to state that there is a great deal of uncertainty in large-scale estimates associated with sand dams. For example, various sources have estimated that there are anywhere from 1500-3000 sand dams in Kenya alone (de Trincheria et al., 2015; de Trincheria et al., 2018; Viducich, 2015). There are few published studies on sand dams, so much of the large-scale information comes from somewhat unsubstantiated claims and estimates. This, in part, motivates our desire to increase the body of knowledge surrounding sand dams.

Best regards, Jessica Eisma and Dr. Venkatesh Merwade

---

## Referee Comment (RC1) · Rolf Hut (Referee) · 29 May 2019

**Review of "Investigating the environmental response to water harvesting structure: a field study in Tanzania"**

Article by J.A. Eisma and V. M. Merwade. Review by R. W. Hut

The authors investigate three sand dams in Tanzania on macroinvertebrate habitat, vegetation, streambank erosion and local water table. Data like this is hard to gather and much needed to evaluate the impact of measures such as sand dams. This dataset is in itself a valuable addition to the literature on sand dams. However, I have major concerns with the conclusions that the authors draw based on their data that, in my opinion, need to be addressed before this work can be published in HESS. My concerns focus firstly on the statistical representativeness of the research and the claims that the authors make on sand dams in general based on the research they did on three individual dams. Secondly, I believe the modelling effort is flawed and the conclusions from the modelling (mainly on evaporation) are inaccurate.

All in all I would either suggest major revisions or ask the authors to split this paper in a data paper and an analysis paper. While the analysis paper would need considerable work, the data paper can be published almost as is and is a valuable addition to the scientific literature on sand dams, in my opinion.

I use P4L24 to point to Page 4, Line 24 of the manuscript.

**Statistical representativeness**

In their introduction the authors start by claiming that previous work "do not tell the whole story of sand dam impacts and this had created a false perception." P2L18. They continue to dismiss previous work as anecdotal and not scientific by stating that "This study aims to respond to anecdotes with science" P2L20. The authors set their goal clearly: "This diversity of features ensures that the sand dams included will be representative of the sand dams found throughout the region, and this study will therefore create a holistic understanding of how a sand dam interacts with the local environment.". Claiming that this can be done be examining only three dams out of the 1500 dams in sub-Saharan Africa [P2L5], even when chosen carefully, grossly underestimates the differences among the dams.
It is the very nature of geoscience in general and land surface studies like hydrology in particular that every locale is different. Isolated experiments in particular locations will never

draw the entire picture of the (luckily) very diverse land surface. In their conclusions the authors contribute dam failure or success on specific attributes of the dams they study. For example: the increase in vegetation was higher at the Soweto dam and the Soweto dam resides in a relatively flat area P11L27. The authors extrapolate this to the conclusion that "to maximize positive impact of a sand dam on local vegetation, sand dams should be build in flat areas." Since there are far more factors that influence success of a dam, some of which the authors touch upon, this is a way too broad statement. The observation that for this particular dam, the local vegetation is positively affected is a valuable observation. The hypothesis that in this particular case that is caused by the relative flatness of the area is a valid hypothesis that future researchers can test if it holds in a broader context.

I want to ask the authors to skim through their manuscript for places where their conclusions and claims extend beyond the data they have gathered and adjust their manuscript to bring conclusions and data in line with each other.

**Modelling efforts and conclusions**

The authors use a water balance model to model how much water the dams are are losing over time. I have several issues here:

1. The model calculates Qout based on the other terms, it therefore accumulates all errors in Qout, including errors because of terms not included in the model
2. I assume from figure 7 that the authors start the dams "full". This is not made explicit in the article.
3. The inflow term 0.038CP(t) accounts (I think, not made clear) for the amount of rain water that falls on the dam itself and is subsequently stored? I would argue that during a rain event all water from upstream would be routed over the stream-bed thus re-filling it. The 0.038 term from Aerts 2007 relates to the total amount of water a sand dam saves from annual discharge to see if dams have an impact on downstream water availability. This factor can not be used as the authors do.
4. The 0.15 factor from Kumar 2018 relates to the percentage of evap that is canopy evap in the Noah LSM, which, if I recall correctly, was not calibrated for the region that the authors use it for. I would guess that on the African regions of interest here, the amount of canopy versus other evap would be different.
5. The Qcomm term is estimated based on conversation with locals. This is understandable given the constraints of the research, but introduces a very large uncertainty. In my own research we observed that some people living close to the dam would, against the deal with the entire community, use a machine pump to irrigate their lands from the sand reservoir, draining the reservoir very fast (Hut 2008).

Based on this concerns with nearly every term of the water balance, I would argue that any conclusions based on the final term Qout, should be taken very carefully. In the conclusions the authors related the decline in water to evaporation only and claim that 400.000 L per week is "lost". I would first ask the authors to convert this to the usual mm/day units to compare if this is

remotely realistic. If I assume that the evap only comes from the sand-reservoir behind the dam and the reservoir is 10 times as long as the dam is wide (25 m), this would mean 400.000/(7 * 0.5 * 25 * 250) is about 18 mm of evap per day which seems unrealistically high. Secondly, I think that unreported withdrawals and seepage have an influence here that the authors don't take into account.

I believe the data on water volume from the measurements are very valuable, but I would ask the authors to have a second look at their model, given all the concerns above

---

## Referee Comment (RC2) · Anonymous Referee #2 · 24 Jun 2019

I agree with the authors' assertions that the literature can paint a rosy picture of sand dams by studying a few successful examples. It is therefore a bit disappointing that only three dams are considered in this study, and also that there is no attempt to understand the communities' perceptions of the sand dams. But nevertheless there is a good dataset, although I disagree with the conclusions.

I am curious as to why FLDAS is selected, there is no justification. It seems to assume that all water loss must be through evaporation rather than considering that there may be leakage from the trapped sand, either under the dam wall or through the river bed. This could help explain some of the results (e.g. p. 8 line 16-18, p. 10 lines 1-4, p. 10

line 10, p. 10 line 20-21. p. 11 line 2, p. 12 line 16, p. 12 line 19) and would have a big impact on the conclusions. The established literature on evaporation is only referenced right at the end of the discussion (p. 12 line 25)

In order to calculate the storage in the sand dam (p.6 line 30-31), why not just assume that it is fully saturated at the end of the wet season? This could be supported if you observe water ponded on top of the trapped sand.

I am curious as to why the WTMWs were the only attempt to measure water levels the sand dams. Piezometers or even excavated holes in the sand dam could have provided a more comprehensive picture. In my experience observing the water depth in scoop holes that the communities dig can be an excellent indicator of overall water levels, but I don't know if there were present here. I am also not surprised by the results of the macroinvertebrate study. That a dry river bed in an arid region contains no macro invertebrates seems hardly to be a surprise. This methodology seems to be more suited to perennial rivers.

There are results on the sediment grain size (p 7 line 19) but no methodology to measure it.

Other comments: p. 8 line 4 – I am struggling to see how the vegetated cover is correlated to the land slope in figure 3. Could this be confirmed through a statistical test? p. 8 line 8-9 – point 1 is poorly explained, and again on p. 11 line 29-32. A figure would help here. p. 9 line 21-25 – this is hard to follow. p. 9 line 24 – how can soil be assessed properly in advance to avoid this type of failure? p.10 line 31 – by "subsurface water reservoir" do you mean the underlying aquifer of the trapped sand? p. 12 line 6 – please be more specific on why the stream channel migration is important.

---

## Author Comment (AC2) · 27 Jun 2019

**Authors' Response**

We would like to thank the reviewer for his constructive comments on the manuscript. We have considered the reviewer's comments and provide the following responses.

**Statistical Representativeness**

This study was designed to investigate the claim frequently made by non-scientific bodies that sand dams "revitalize the entire ecosystem." This is a claim sometimes repeated, although to a lesser extent, in the introductory sections of sand dam journal articles, but the current body of literature has not tested nor necessarily supported this claim. We do not intend to dismiss the existing sand dam work of various researchers. Rather, we want to challenge the unverified claims about sand dams made by invested parties, primarily NGOs. In having built our study on the foundations laid by the handful of published sand dam studies, we recognize and value the contributions of prior studies. We have altered the language of our primary objective so as to make clear that we are not dismissing the scientific work published to date but rather investigating the claims made by nonscientific bodies.

We thank the reviewer for drawing our attention to the broad conclusions in the manuscript that were not adequately supported by the discussion and/or strength of the data. In some instances, we agree that the language should be softened to ensure that we do not make claims that cannot be fully backed by the literature and statistical representativeness of the data. We will include additional discussion in the manuscript to support some of the conclusions that we believe are justified and adjust other conclusions to ensure they align with the representativeness of the data collected.

**Modelling Efforts and Conclusions**

An important distinction must be made concerning the motivation for developing a water balance model. The field data of the groundwater levels around the sand dams provide how much water the sand dams are losing over time. The model is being employed to inform the estimate of the various *causes* of those losses. This distinction will be clarified in the article before publication. The reviewer's issues with the model are addressed by number.

1. **The model calculates Qout based on the other terms, it therefore accumulates all errors in Qout, including errors because of terms not included in the model**
   There is uncertainty in the model, because the water balance is a simplified representation and the forcing data (FLDAS) is largely modelled data itself. Despite the uncertainty, the authors are confident that the relative magnitude of the terms in the model is reliable. The relative magnitude of the terms is the primary focus of the conclusions drawn from the model. We will add an explicit statement noting the uncertainty in the model.
2. **I assume from figure 7 that the authors start the dams "full". This is not made explicit in the article.**
   The analysis displayed in Figure 7 begins in the middle of the rainy season, so it is assumed that the sand dams are full at this point. However, this will change as a result of our model changes resulting from point 3, below. A sentence indicating the initial condition of the model will be added to the paper.
3. **The inflow term 0.038CP(t) accounts (I think, not made clear) for the amount of rain water that falls on the dam itself and is subsequently stored? I would argue that during a rain event all water from upstream would be routed over the stream-bed thus re-filling it. The 0.038 term from Aerts 2007 relates to the total amount of water a sand dam saves from annual**

**discharge to see if dams have an impact on downstream water availability. This factor cannot be used as the authors do.**

You are correct. The 0.038 term, hereafter capture ratio, from Aerts (2007) is the maximum proportion of annual discharge that a sand dam is expected to capture and store. In the water balance model proposed in the manuscript, the inflow term, 0.038CP(t), accounts only for the runoff that is expected to occur from the study areas indicated in manuscript Figure 1c,d. The area included in the model, therefore, is greater than the dam itself, but smaller than the upstream watershed. The watershed upstream of the Chididimo sand dam is 3.3 km$^2$ and relatively uniform. This allows the watershed to be modelled relatively well using the rational method for overland flow with a capture ratio of 0.038. The watershed upstream of the Soweto sand dam, however, is 262.1 km$^2$ and includes commercial farmland and an 18.6 km$^2$ wetland area. The Soweto sand dam is much too small to capture 0.038 of the runoff generated by such a large watershed. To more accurately represent the volume of water captured by the Soweto sand dam, the capture ratio would need to be reduced to around 0.00025 and adjusted for seasonal variability. However, such a methodology seems somewhat speculative, and we would prefer to be consistent in our methodology. In summary, the inflow term proposed in the water balance model is insufficient for accurately representing the volume of water captured by the Soweto sand dam. To address this issue, we initialized the water balance at the week of last rainfall. Therefore, there are no inflows to the theoretical model. The model now solely describes the loss factors, which is our primary interest (see Eq. 1 and Fig. 3, below).

4. **The 0.15 factor from Kumar 2018 relates to the percentage of evap that is canopy evap in the Noah LSM, which, if I recall correctly, was not calibrated for the region that the authors use it for. I would guess that on the African regions of interest here, the amount of canopy versus other evap would be different.**

   While the Noah LSM may not have been calibrated for East Africa, there are examples of Noah LSM being used over East Africa (Anderson et al., 2012; Yilmaz et al., 2014). Further, the evapotranspiration (ET) data used in the theoretical model presented in this paper is from FLDAS. The iteration of FLDAS used was developed based on the Noah LSM, but is specifically designed for use in sub-Saharan Africa (McNally et al., 2017). Furthermore, we believe the 0.15 factor for ET partitioning described in Kumar et al. (2018) to be appropriate. The climate in Dodoma, Tanzania is classified as hot semi-arid, which is also the climate in parts of the southwestern US and northern Mexico. From the figure below, included in the Kumar et al. (2018) paper, you can see that the canopy ET partition fraction for much of the southwestern US and northern Mexico falls between 0.1 and 0.2.

[Figure]

**Figure 1:** Mean of the $\mathrm{ET}$ partition fraction of canopy ET (unitless; from Kumar et al., 2018).

We recognize that the 0.15 ET partition fraction may not be perfectly accurate, but a study such as Kumar et al. (2018) has not been performed for East Africa. There is no better estimate available.

To more accurately represent the amount of water lost to ET within the control volumes, the ET will be multiplied by a factor of 0.85 for the area within the sand dams and will be multiplied by a factor of 1 outside of the sand dams. There is potential canopy ET outside the sand dams but within the control volume.

In addition, we realize that our manuscript is not clear on the size of the control volume to which the theoretical model is applied. Figure 2, below, provides the control volume for the Chididimo and Soweto sand dams. The control volume includes the sand dam and all area enclosed by the water table monitoring wells (WTMW) installed around the study area. Figure 2 will not be added to the manuscript, but a statement clarifying the extent of the control volumes will be added.

[Figure]

**Figure 2:** Height of subsurface water around the (a) Chididimo sand dam and the (b) Soweto sand dam.

5. **The Qcomm term is estimated based on conversation with locals. This is understandable given the constraints of the research, but introduces a very large uncertainty. In my own research we observed that some people living close to the dam would, against the deal with the entire community, use a machine pump to irrigate their lands from the sand reservoir, draining the reservoir very fast (Hut 2008).**
We appreciate and understand the concern regarding the uncertainty of the community withdrawals variable. We will add a sentence to the manuscript indicating that the estimate of community withdrawals has an unknown degree of uncertainty. We, however, have no reason to believe that the community members were engaging in machine pumping of the water. The Soweto sand dam did have many areas under cultivation near the dam, but we did not see any evidence of machine pumping. Also, the community water group was very strict with its members regarding withdrawals under the guidance of the local chairman, including such measures as locking access to the hand pump. The Chididimo sand dam was much more difficult to access, and only had one small area nearby under cultivation. These reasons coupled with the lack of evidence lead us to believe that machine pumping was not a significant factor in the rate of water loss in the Soweto and Chididimo sand dams.

Given the above, Eq. (1) will be modified to:

$$Q_{out,dry\ season}(t) = -\alpha \times E(t) - Q_{sb}(t) - Q_{com}(t) \qquad (1)$$

Where $Q_{out,dry\ season}$ is the rate of water loss from the sand dam after the end of the rainy season, $E(t)$ is total evapotranspiration modified by $\alpha$, which is 0.85 for the area within the sand dam and 1 for the area outside of the sand dam, $Q_{sb}$ is baseflow-groundwater runoff, and $Q_{com}$ is the community's water use. Eq. (1) is integrated over time and subtracted from the volume of water in the control volume at the end of the rainy season to create a theoretical volume of water curve for the sand dam area.

Given the changes to Eq. (1), Figure 7 is modified to:

[Figure]

**Figure 3:** Volume of water in the area enclosed by the WTMWs of the (a) Chididimo and (b) Soweto sand dams. The field data line shows the volume of water in the study area during the specified week. The theoretical line, initiated at the end of the rainy season, shows the theoretical amount of water in the study areas, calculated by integrating Eq. (1). The theoretical line accounts for losses due to evapotranspiration, baseflow-groundwater runoff, and community use.

The changes to Eq. (1) result in the following loss partitioning (summary of changes, not for inclusion in manuscript):

**Table 1:** Sand dam stored water loss partitioning during the dry season

| Loss Partition (%) | Chididimo | | Soweto | |
|---|---|---|---|---|
| | Old Eq. (1) | Updated Eq. (1) | Old Eq. (1) | Updated Eq. (1) |
| Evapotranspiration | 85 | 53 | 51 | 35 |
| Baseflow-groundwater runoff | 1 | 1 | 1 | 1 |
| Community use | 5 | 4 | 8 | 4 |
| Seepage | - | - | - | 25 |
| Unaccounted | 9 | 42 | 40 | 35 |

From Table 1, above, you can see that the loss partitioning for the Chididimo sand dam did not change much as a result of updating Eq. (1), with the exception of the unaccounted fraction which is still understood to be primarily ET losses. The loss partitioning for the Soweto sand dam did change significantly with the

inclusion of seepage losses. The Soweto sand dam does lose water as a result of seepage, as evidenced by the scoopholes community members dig just downstream of the dam from which they collect water. Community members did not exhibit the same behavior at the Chididimo sand dam, therefore we do not believe that there is significant seepage occurring at the Chididimo sand dam. At Soweto, the community members expressed an inability to abstract water from the sand dam after approximately the 30[th] week of the year. Therefore, we believe that the water lost from the Soweto sand dam after the 30[th] week is likely due primarily to seepage losses. Assuming seepage is relatively constant, we can extrapolate this portion of the plot back to the end of the rainy season and get an estimate of total seepage losses from the Soweto sand dam during the dry season (Fig. 3b). There is likely minimal ET loss occurring after the 30[th] week, because the water is deep underground at that point (Hellwig, 1973).

Figure 4 shows partitioning changes in the water lost from the study areas during the dry season. This figure will be added to the manuscript with additional explanatory text.

[Figure]

**Figure 4:** Fractional causes of water lost from the area enclosed by the WTMWs of the (a) Chididimo and (b) Soweto sand dams.

In regards to your estimate of the mm/day rate of evapotranspiration from the sand dams, the estimate is quite high due to the misunderstanding about the size of the control volume from which the field data line is determined for Figure 3. From Figure 2, above, there is clearly a great deal of seepage from the sand dam through the streambanks, so the area from which evapotranspiration is occurring is significantly greater than simply the surface area of the sand dams. The updated estimates for the rate of evapotranspiration losses from the Chididimo and Soweto sand dams are: 380 000 L/week and 1 117 000 L/week (slope of total loss-seepage loss, Fig. 3), respectively. With this understanding, the rate of evapotranspiration losses can be calculated as follows:

$$Total\ Loss\ Rate(L_T)\frac{mm}{day} = \frac{Slope\ of\ Total\ Loss\ Line\ \frac{L}{week}}{7\frac{days}{week} \times Control\ Area\ (m^2)}$$

$$Avg.\ Evap\ Rate\ \left(\frac{mm}{day}\right) = Total\ Loss\ Rate\frac{mm}{day} \times (Evap + Unaccounted\ Fraction)$$

$$Avg.\,Evap\,Rate_{Chididimo,shallow}\left(\frac{mm}{day}\right)=\frac{380\,000\,\frac{L}{week}\times(0.52+0.42)}{7\,\frac{days}{week}\times(163\,m\times198\,m)}=1.56\,\frac{mm}{day}$$

$$Avg.\,Evap\,Rate_{Soweto,shallow}\left(\frac{mm}{day}\right)=\frac{1\,300\,000\,\frac{L}{week}\times(0.35+0.35)}{7\,\frac{days}{week}\times(227\,m\times185\,m)}=3.09\,\frac{mm}{day}$$

The above evapotranspiration rates are in general agreement with the sand dam sub-surface evaporation rate of 2.4 mm/day found by Borst and de Haas (2006). It should also be noted that the Soweto estimate is valid only for the rate of ET when the sand dam is relatively full. As is clear in Figs. 3b and 4b, the rate of ET decreases as the volume of water in the sand dam decreases.

**Additional References**

Anderson, W. B., Zaitchik, B.F., Hain, C.R., Anderson, M.C., Yilmaz, M.T., Mecikalski, J., and Schultz, L.: Towards an integrated soil moisture drought monitor for East Africa. Hydrol. Earth Syst. Sci., 16, 2893–2913, doi:10.5194/hess-16-2893-2012, 2012.

Yilmaz, M. T., Anderson, M.C., Zaitchik, B., Hain, C.R., Crow, W.T., Ozdogan, M., Chun, J.A., and Evans, J.: Comparison of prognostic and diagnostic surface flux modeling approaches over the Nile River basin. Water Resour. Res., 50(1), 386–408, doi:10.1002/2013WR014194, 2014.

---

## Author Comment (AC3) · 30 Jun 2019

**Authors' Response**

We would like to thank the reviewer for the constructive comments on the manuscript. We have considered the reviewer's comments and provide the following responses.

*It is therefore a bit disappointing that only three dams are considered in this study, and also that there is no attempt to understand the communities' perceptions of the sand dams.*

This study's success relied on active participation of community water groups to help collect long-term datasets. Of the 15 sand dams in Tanzania known to us, only three still have active community water groups maintaining the dams. This, in addition to funding limitations and long travel times, prevented us from performing in-depth field studies for more than three sand dams.

We worked very closely with the community water groups over the course of the study and developed a good understanding of their perception of their sand dam. However, the social aspect of sand dams is not the focus of this study, so any related commentary has been omitted.

*I am curious as to why FLDAS is selected, there is no justification.*

FLDAS was selected as a proxy for climate data, because there is not a consistently reliable source of climate data available for Dodoma or Longido, Tanzania. FLDAS is not the perfect substitute, but it has been specifically designed and validated for use in sub-Saharan Africa. A line will be added to the manuscript explaining the choice of the FLDAS dataset.

*It seems to assume that all water loss must be through evaporation rather than considering that there may be leakage from the trapped sand, either under the dam wall or through the riverbed. This could help explain some of the results (e.g. p. 8 line 16-18, p. 10 lines 1-4, p. 10 line 10, p. 10 line 20-21. p. 11 line 2, p. 12 line 16, p. 12 line 19) and would have a big impact on the conclusions. The established literature on evaporation is only referenced right at the end of the discussion (p. 12 line 25).*

Based on the relative magnitude of loss terms in Eq. (1) in the manuscript and the prolonged availability of near-surface water resulting from storage in the sand dam, we believe that most of the unaccounted losses are due to evapotranspiration at the Chididimo sand dam. However, we have considered the comments of Reviewer 1, and updated our conclusions regarding water loss at the Soweto sand dam. Unlike the Chididimo sand dam, the Soweto sand dam did exhibit signs of seepage occurring under the dam wall. We will add a statement to the manuscript acknowledging that seepage through the streambed could be contributing to the unaccounted water loss.

*In order to calculate the storage in the sand dam (p.6 line 30-31), why not just assume that it is fully saturated at the end of the wet season? This could be supported if you observe water ponded on top of the trapped sand.*

Thank you for your suggestion. In response to comments from Reviewer 1, we have updated Eq. (1) and initiated the theoretical water loss model at the end of the rainy season. This will be updated in the manuscript.

*I am curious as to why the WTMWs were the only attempt to measure water levels the sand dams. Piezometers or even excavated holes in the sand dam could have provided a more comprehensive picture. In my experience observing the water depth in scoop holes that the communities dig can be an excellent indicator of overall water levels, but I don't know if there were present here.*

Borst and de Haas (2006) installed piezometers in the sand dam itself, so we did not think it necessary to repeat this arrangement. Instead, we wanted to focus on how the sand dam affects the water table outside of the stream channel, since the idea that sand dams raise the local groundwater table has really only been explored via modelling. Therefore, we installed WTMWs in the streambanks and the surrounding area to track how the water table was changing over time. We did not consider tracking water levels in the scoop holes dug by community members but will consider this as a viable methodology for future studies.

*I am also not surprised by the results of the macroinvertebrate study. That a dry river bed in an arid region contains no macro invertebrates seems hardly to be a surprise. This methodology seems to be more suited to perennial rivers.*

Boulton et al. (1992), Stubbington et al. (2009), and Verdonschot et al. (2014) all successfully used variations of the methodology described in the manuscript to sample dry river beds. Boulton et al. (1992) even sampled an ephemeral river in the Arizona desert. We believe that we used the appropriate methodology for collecting macroinvertebrates from an ephemeral streambed. If sand dams were less homogeneous and therefore more suitable habitats for macroinvertebrates, we believe we would have been successful in our sampling attempts.

Macroinvertebrate sampling in perennial rivers is most often conducted by placing a net, typically a surber sampler, facing upstream on the streambed and then disturbing the upstream streambed to release any macroinvertebrates nestled there. The flowing stream then carries the macroinvertebrates into the net, where they are captured and identified.

*There are results on the sediment grain size (p 7 line 19) but no methodology to measure it.*

A formal sediment grain size analysis was not performed. The determination that the Chididimo and Soweto sand dams consist primarily of fine- and coarse-grained sand was based on a visual and tactile assessment of the material. A statement will be added to the manuscript to clarify this point.

*p. 8 line 4 – I am struggling to see how the vegetated cover is correlated to the land slope in figure 3. Could this be confirmed through a statistical test?*

The paragraph immediately preceding (p. 7 line 25-p.8 line 2) discusses the relationship between land slope and vegetated cover in the context of Figure 3. This relationship is not immediately apparent from the figure alone. The statement on p. 8 line 4 will be edited to reduce the chance for reader confusion.

We will look into performing an appropriate statistical test to validate our discussion of the relationship between land slope and vegetated cover.

*p. 8 line 8-9 – point 1 is poorly explained, and again on p. 11 line 29-32. A figure would help here.*

Figure 1, below, will be added to the manuscript to help clarify our point. In addition, further discussion of how flat land can generally support more vegetation than steeply sloping land will be added to the manuscript. Storm surface runoff travels more slowly over flat land than steeply sloping land, allowing more time for water to infiltrate into the ground. The infiltrated water recharges the groundwater and increases the soil moisture, which supports vegetation growth. In addition, when a sand dam creates a locally raised water table in an area with flat land, the water table will be nearer to the ground surface, and therefore plant roots, than if the sand dam were surrounded by steeply sloping land (see Fig. 1, below). The water table has a positive impact on the soil moisture of the unsaturated soil layer just above the water table, and this additional moisture supports higher rates of vegetation growth. Shemsaga et al. (2018) will be

added to the manuscript in support of using healthy vegetated cover as an indication of groundwater in Dodoma.

[Figure]

**Figure 1: The roots of plants growing on a (a) steep slope will be farther from the locally raised water table created by a sand dam, and therefore have less access to soil water, than vegetation growing on a (b) gentle slope.**

*p. 9 line 21-25 – this is hard to follow.*

These lines will be updated to remove unnecessary detail and clarify the point.

*p. 9 line 24 – how can soil be assessed properly in advance to avoid this type of failure?*

To our knowledge, this point is not well-discussed in the available literature. However, it is our understanding that the soil composition of the streambed before a sand dam is constructed can be helpful in determining the distribution of grain sizes likely to be captured by a constructed sand dam. This information coupled with knowledge of the sediment load typically carried by the stream when flowing can inform one's decision regarding the necessity of constructing the spillway of a sand dam in stages. A statement with this information will be added to the manuscript.

*p.10 line 31 – by "subsurface water reservoir" do you mean the underlying aquifer of the trapped sand?*

Yes, we mean the water stored between the grains of sand in the sand dam. The wording of this statement will be altered to reduce reader confusion.

*p. 12 line 6 – please be more specific on why the stream channel migration is important.*

Noted. We will add a few sentences to the manuscript explaining how increased erosion and stream channel migration can lead to the failure of a sand dam.

**Additional References**

Shemsanga, C., Muzuka, A.N.N., Martz, L., Komakech, H., and Mcharo, E.: Indigenous knowledge on development and management of shallow dug wells of Dodoma Municipality in Tanzania, Appl. Water Sci. 8, 59, https://doi.org/10.1007/s13201-018-0697-7, 2018.

---

## Author Comment (AC4) · 8 Jul 2019

**Response to Reviewer 1**

We would like to thank the reviewer for his constructive comments on the manuscript. We have considered the reviewer's comments and provide the following responses.

**Statistical Representativeness**

This study was designed to investigate the claim frequently made by non-scientific bodies that sand dams revitalize the entire ecosystem. This is a claim sometimes repeated, although to a lesser extent, in the introductory sections of sand dam journal articles, but the current body of literature has not tested nor necessarily supported this claim. We do not intend to dismiss the existing sand dam work of various researchers. Rather, we want to challenge the unverified claims about sand dams made by invested parties, primarily NGOs. In having built our study on the foundations laid by the handful of published sand dam studies, we recognize and value the contributions of prior studies. We have altered the language of our primary objective so as to make clear that we are not dismissing the scientific work published to date but rather investigating the claims made by nonscientific bodies.

*P2L5-6 has been updated to read: This has been the case with sand dams in sub-Saharan Africa. Sub-Saharan Africa is home to over 3000 sand dams, yet approximately 50 % of sand dams are essentially non-functioning (de Trincheria et al., 2018; Viducich, 2015).* [A recent publication, de Trincheria et al. (2018) corroborates Viducich's (2015) claim that there are 3000 sand dams, not 1500 sand dams in sub-Saharan Africa.]

*P2L20-21 has been updated to read: This study examines claims made by non-scientific bodies about sand dam impacts by investigating how diverse sand dams influence macroinvertebrate habitat, vegetation, erosion, and groundwater recharge in the riparian zone.*

*P2L24-25 has an additional statement: Answering these questions will provide some insight into the validity of the claim that sand dams revitalize the entire ecosystem (Reversing Land Degradation and Desertification, n.d.; Sand Dams, n.d.).*

*P2L28-29 has been updated to read: This diversity of features provides a broad representation of the sand dams found throughout the region, and this study will therefore create a better understanding of how a sand dam interacts with the local environment.*

We thank the reviewer for drawing our attention to the broad conclusions in the manuscript that were not adequately supported by the discussion and/or strength of the data. In some instances, we agree that the language should be softened to ensure that we do not make claims that cannot be fully backed by the literature and statistical representativeness of the data. We will include additional discussion in the manuscript to support some of the conclusions that we believe are justified and adjust other conclusions to ensure they align with the representativeness of the data collected.

*P13L10-12 have been updated to read: Holding all other variables constant, building a sand dam in a flat area would likely maximize the positive impact of the sand dam on local vegetation. However, this needs to be further explored to see if the trend holds when more sand dams are examined.*

**Modelling Efforts and Conclusions**

An important distinction must be made concerning the motivation for developing a water balance model. The field data of the groundwater levels around the sand dams provide how much water the sand dams are losing over time. The model is being employed to inform the estimate of the various *causes* of those losses. This distinction was clarified in the article.

*P6L24 has been updated to read: To help determine the various causes of water loss from the sand dam and their relative magnitude, a water balance was calculated …*

The reviewer's issues with the model are addressed by number.

1. **The model calculates Qout based on the other terms, it therefore accumulates all errors in Qout, including errors because of terms not included in the model**
   There is uncertainty in the model, because the water balance is a simplified representation and the forcing data (FLDAS) is largely modelled data itself. Despite the uncertainty, the authors are confident that the relative magnitude of the terms in the model is reliable. The relative magnitude of the terms is the primary focus of the conclusions drawn from the model. We added an explicit statement noting the uncertainty in the model.

   *P7L1-4 has an additional statement: The theoretical volume of water resulting from Eq. (1) has a high degree of uncertainty, because it is a simplified representation of water loss that utilizes modelled FLDAS data. However, the relative magnitude of the loss terms is likely reliable, and this is the primary focus of the conclusions that will be drawn from the model.*

2. **I assume from figure 7 that the authors start the dams "full". This is not made explicit in the article.**
   The analysis displayed in Figure 7 begins in the middle of the rainy season, so it is assumed that the sand dams are full at this point. However, this will change as a result of our model changes resulting from point 3, below. A sentence indicating the initial condition of the model was added to the manuscript.

   *P6L29 has been updated to read: $Q_{out,dry\ season}$ is the rate of water loss from the sand dam after the end of the rainy season…*

   *P6L31-P6L33 has been updated to read: Eq. (1) is integrated over time and subtracted from the volume of water in the control volume at the end of the rainy season to create a theoretical volume of water curve for the sand dam area.*

   *Figure 7 (now Figure 8) caption has been updated to read: The theoretical line, initiated at the end of the rainy season, shows the cumulative theoretical amount of water in the study area…*

3. **The inflow term 0.038CP(t) accounts (I think, not made clear) for the amount of rain water that falls on the dam itself and is subsequently stored? I would argue that during a rain event all water from upstream would be routed over the stream-bed thus re-filling it. The 0.038 term from Aerts 2007 relates to the total amount of water a sand dam saves from annual discharge to see if dams have an impact on downstream water availability. This factor cannot be used as the authors do.**
   You are correct. The 0.038 term, hereafter capture ratio, from Aerts (2007) is the maximum proportion of annual discharge that a sand dam is expected to capture and store. In the water balance model proposed in the manuscript, the inflow term, 0.038CP(t), accounts only for the runoff that is expected to occur from the study areas indicated in manuscript Figure 1c,d. The area included in the model, therefore, is greater than the dam itself, but smaller than the upstream watershed. The watershed upstream of the Chididimo sand dam is 3.3 km$^2$ and relatively uniform. This allows the watershed to be modelled relatively well using the rational method for overland flow with a capture

ratio of 0.038. The watershed upstream of the Soweto sand dam, however, is 262.1 km$^2$ and includes commercial farmland and an 18.6 km$^2$ wetland area. The Soweto sand dam is much too small to capture 0.038 of the runoff generated by such a large watershed. To more accurately represent the volume of water captured by the Soweto sand dam, the capture ratio would need to be reduced to around 0.00025 and adjusted for seasonal variability. However, such a methodology seems somewhat speculative, and we would prefer to be consistent in our methodology. In summary, the inflow term proposed in the water balance model is insufficient for accurately representing the volume of water captured by the Soweto sand dam. To address this issue, we initialized the water balance at the week of last rainfall. Therefore, there are no inflows to the theoretical model. The model now solely describes the loss factors, which is our primary interest (see Eq. 1 and Fig. 3, below).

4. **The 0.15 factor from Kumar 2018 relates to the percentage of evap that is canopy evap in the Noah LSM, which, if I recall correctly, was not calibrated for the region that the authors use it for. I would guess that on the African regions of interest here, the amount of canopy versus other evap would be different.**

    While the Noah LSM may not have been calibrated for East Africa, there are examples of Noah LSM being used over East Africa (Anderson et al., 2012; Yilmaz et al., 2014). Further, the evapotranspiration (ET) data used in the theoretical model presented in this paper is from FLDAS. The iteration of FLDAS used was developed based on the Noah LSM, but is specifically designed for use in sub-Saharan Africa (McNally et al., 2017). Furthermore, we believe the 0.15 factor for ET partitioning described in Kumar et al. (2018) to be appropriate. The climate in Dodoma, Tanzania is classified as hot semi-arid, which is also the climate in parts of the southwestern US and northern Mexico. From the figure below, included in the Kumar et al. (2018) paper, you can see that the canopy ET partition fraction for much of the southwestern US and northern Mexico falls between 0.1 and 0.2.

[Figure]

**Figure 1: Mean of the ET partition fraction of canopy ET (unitless; from Kumar et al., 2018).**

We recognize that the 0.15 ET partition fraction may not be perfectly accurate, but a study such as Kumar et al. (2018) has not been performed for East Africa. There is no better estimate available.

To more accurately represent the amount of water lost to ET within the control volumes, the ET will be multiplied by a factor of 0.85 for the area within the sand dams and will be multiplied by a factor of 1 outside of the sand dams. There is potential canopy ET outside the sand dams but within the control volume.

*P6L29-30 has been updated to read: E(t) is total evapotranspiration modified by α, which is 0.85 for the area within the sand dam and 1.00 for the area outside of the sand dam.*

*P7L10-11 has been updated to read: Therefore, total evapotranspiration is reduced by 15 % in Eq. (1) for the portion of the control volume that is occupied by the sand dam.*

In addition, we realize that our manuscript is not clear on the size of the control volume to which the theoretical model is applied. Figure 2, below, provides the control volume for the Chididimo and Soweto sand dams. The control volume includes the sand dam and all area enclosed by the water table monitoring wells (WTMW) installed around the study area. Figure 2 was not added to the manuscript, but a statement clarifying the extent of the control volumes was added.

[Figure]

Figure 2: Height of subsurface water around the (a) Chididimo sand dam and the (b) Soweto sand dam.

*P6L33-P7L1 has an additional statement: The control volume to which Eq. (1) is applied is the portion of the study area that is enclosed by the installed WTMWs (see Fig. 1 c,d). For Chididimo, this area is 32 274 m², while it is 41 995 m² for Soweto.*

5. **The Qcomm term is estimated based on conversation with locals. This is understandable given the constraints of the research, but introduces a very large uncertainty. In my own research we observed that some people living close to the dam would, against the deal with the entire community, use a machine pump to irrigate their lands from the sand reservoir, draining the reservoir very fast (Hut 2008).**
We appreciate and understand the concern regarding the uncertainty of the community withdrawals variable. We added a sentence to the manuscript indicating that the estimate of community withdrawals has an unknown degree of uncertainty. We, however, have no reason to believe that the community members were engaging in machine pumping of the water. The Soweto sand dam did have many areas under cultivation near the dam, but we did not see any evidence of machine pumping. Also, the community water group was very strict with its members regarding withdrawals under the guidance of the local chairman, including such measures as locking access to the hand pump. The Chididimo sand dam was much more difficult to access, and only had one small area nearby under cultivation. These reasons coupled with the lack of evidence lead us to believe that machine pumping was not a significant factor in the rate of water loss in the Soweto and Chididimo sand dams.

*P7L12-15 has an additional statement: The estimate of the community's use of water from the sand dam has an unknown degree of uncertainty. At least one sand dam researcher has noted that unsanctioned machine pumping of water from sand dams can cause rapid drawdown of stored water (Hut et al., 2008). However, no evidence was present at either Dodoma site to indicate the community was drawing significantly more water from the sand dams than they indicated.*

Given the above, Eq. (1) was modified to:

*P6L28-33:*

$$Q_{out,dry\ season}(t) = -\alpha \times E(t) - Q_{sb}(t) - Q_{com}(t) \,, \tag{1}$$

*where $Q_{out,dry\ season}$ is the rate of water loss from the sand dam after the end of the rainy season, $E(t)$ is total evapotranspiration modified by $\alpha$, which is 0.85 for the area within the sand dam and 1.00 for the area outside of the sand dam, $Q_{sb}$ is baseflow-groundwater runoff, and $Q_{com}$ is the community's water use. Eq. (1) is integrated over time and subtracted from the volume of water in the control volume at the end of the rainy season to create a theoretical volume of water curve for the sand dam area.*

Given the changes to Eq. (1), Figure 7 (now Figure 8 in manuscript) was modified to:

[Figure]

**Figure 3: Volume of water in the area enclosed by the WTMWs of the (a) Chididimo and (b) Soweto sand dams. The field data line shows the volume of water in the study area during the specified week. The theoretical line, initiated at the end of the rainy season, shows the theoretical amount of water in the study areas, calculated by integrating equation (1). The theoretical line accounts for losses due to evapotranspiration, baseflow-groundwater runoff, and community use.**

The changes to Eq. (1) result in the following loss partitioning (summary of changes, not for inclusion in manuscript):

**Table 1:** Sand dam stored water loss partitioning during the dry season

| Loss Partition (%) | Chididimo | | Soweto | |
| --- | --- | --- | --- | --- |
| | Old Eq. (1) | Updated Eq. (1) | Old Eq. (1) | Updated Eq. (1) |
| Evapotranspiration | 85 | 53 | 51 | 35 |

| | | | | |
|---|---|---|---|---|
| Baseflow-groundwater runoff | 1 | 1 | 1 | 1 |
| Community use | 5 | 4 | 8 | 4 |
| Seepage | - | - | - | 25 |
| Unaccounted | 9 | 42 | 40 | 35 |

From Table 1, above, you can see that the loss partitioning for the Chididimo sand dam did not change much as a result of updating Eq. (1), with the exception of the unaccounted fraction which is still understood to be primarily ET losses. The loss partitioning for the Soweto sand dam did change significantly with the inclusion of seepage losses. The Soweto sand dam does lose water as a result of seepage, as evidenced by the scoopholes community members dig just downstream of the dam from which they collect water. Community members did not exhibit the same behavior at the Chididimo sand dam, therefore we do not believe that there is significant seepage occurring at the Chididimo sand dam. At Soweto, the community members expressed an inability to abstract water from the sand dam after approximately the 30[th] week of the year. Therefore, we believe that the water lost from the Soweto sand dam after the 30[th] week is likely due primarily to seepage losses. Assuming seepage is relatively constant, we can extrapolate this portion of the plot back to the end of the rainy season and get an estimate of total seepage losses from the Soweto sand dam during the dry season (Fig. 3b). There is likely minimal ET loss occurring after the 30[th] week, because the water is deep underground at that point (Hellwig, 1973).

*P11L31-P12L5 now reads: The Soweto dam exhibits three distinct phases of water loss: shallow ET, deep ET, and minimal ET (see Fig. 8b). The minimal ET phase occurs during the period in which the community water group indicated they were no longer able to abstract water from the sand dam. At this point, the water table has retreated too far underground for the community to draw water and, at this depth, the rate of ET is likely negligible (Hellwig, 1973). Therefore, most of the water lost during the minimal ET phase is lost due to seepage under the dam wall and/or through the streambed. Unlike Chididimo, the Soweto sand dam does exhibit evidence of seepage—the community members dig scoopholes just downstream of the dam from which they collect water. The seepage loss at Soweto occurs at a rate of approximately 0.4 mm day[-1] and accounts for 25 % of the water stored by the sand dam at the end of the rainy season, leaving only 35 % of Soweto's water loss unaccounted. The seepage rate essentially remains constant throughout the shallow and deep ET phases (see Fig. 8b).*

In regards to your estimate of the mm/day rate of evapotranspiration from the sand dams, the estimate is quite high due to the misunderstanding about the size of the control volume from which the field data line is determined for Figure 3. From Figure 2, above, there is clearly a great deal of seepage from the sand dam through the streambanks, so the area from which evapotranspiration is occurring is significantly greater than simply the surface area of the sand dams. The updated estimates for the rate of evapotranspiration losses from the Chididimo and Soweto sand dams are: 380 000 L/week and 1 117 000 L/week (slope of total loss-seepage loss, Fig. 3), respectively. With this understanding, the rate of evapotranspiration losses can be calculated as follows:

$$Total\ Loss\ Rate(L_T)\frac{mm}{day} = \frac{Slope\ of\ Total\ Loss\ Line\ \frac{L}{week}}{7\frac{days}{week} \times Control\ Area\ (m^2)}$$

$$Avg.\ Evap\ Rate\ \left(\frac{mm}{day}\right) = Total\ Loss\ Rate\frac{mm}{day} \times (Evap + Unaccounted\ Fraction)$$

$$Avg.\,Evap\,Rate_{Chididimo,shallow}\left(\frac{mm}{day}\right) = \frac{380\,000\frac{L}{week} \times (0.53 + 0.42)}{7\frac{days}{week} \times (163\,m \times 198\,m)} = 1.56\frac{mm}{day}$$

$$Avg.\,Evap\,Rate_{Soweto,shallow}\left(\frac{mm}{day}\right) = \frac{1\,300\,000\frac{L}{week} \times (0.35 + 0.35)}{7\frac{days}{week} \times (227\,m \times 185\,m)} = 3.09\frac{mm}{day}$$

The above evapotranspiration rates are in general agreement with the sand dam sub-surface evaporation rate of 2.4 mm/day found by Borst and de Haas (2006). It should also be noted that the Soweto estimate is valid only for the rate of ET when the sand dam is relatively full. As is clear in Figs. 3b and 4b, the rate of ET decreases as the volume of water in the sand dam decreases.

**Response to Reviewer 2**

We would like to thank the reviewer for the constructive comments on the manuscript. We have considered the reviewer's comments and provide the following responses.

**It is therefore a bit disappointing that only three dams are considered in this study, and also that there is no attempt to understand the communities' perceptions of the sand dams.**

This study's success relied on active participation of community water groups to help collect long-term datasets. Of the 15 sand dams in Tanzania known to us, only three still have active community water groups maintaining the dams. This, in addition to funding limitations and long travel times, prevented us from performing in-depth field studies for more than three sand dams.

We worked very closely with the community water groups over the course of the study and developed a good understanding of their perception of their sand dam. From day one, the communities expressed frustration that the sand dams did not serve as a water source throughout the dry season. One community, Soweto, went so far as to lock the handpump at the sand dam to limit easy access to the water. However, the social aspect of sand dams is not the focus of this study, so any related commentary has been omitted.

**I am curious as to why FLDAS is selected, there is no justification.**

FLDAS was selected as a proxy for climate data, because there is not a consistently reliable source of climate data available for Dodoma or Longido, Tanzania. FLDAS is not the perfect substitute, but it has been specifically designed and validated for use in sub-Saharan Africa. A line was added to the manuscript explaining the choice of the FLDAS dataset.

*P6L25-26 now contains the following statement: FLDAS data was used as a proxy for climate data, because there is not a reliable source of climate data freely available for Dodoma, Tanzania.*

**It seems to assume that all water loss must be through evaporation rather than considering that there may be leakage from the trapped sand, either under the dam wall or through the riverbed. This could help explain some of the results (e.g. p. 8 line 16-18, p. 10 lines 1-4, p. 10 line 10, p. 10 line 20-21. p. 11 line 2, p. 12 line 16, p. 12 line 19) and would have a big impact on the conclusions. The established literature on evaporation is only referenced right at the end of the discussion (p. 12 line 25).**

Based on the relative magnitude of loss terms in Eq. (1) in the manuscript and the prolonged availability of near-surface water resulting from storage in the sand dam, we believe that most of the unaccounted losses are due to evapotranspiration at the Chididimo sand dam. However, we have considered the comments of Reviewer 1, and updated our conclusions regarding water loss at the Soweto sand dam. Unlike the

Chididimo sand dam, the Soweto sand dam did exhibit signs of seepage under the dam wall. We added a statement to the manuscript acknowledging that seepage through the streambed could be contributing to the unaccounted water loss.

*P11L11-12 has been updated to read: The unaccounted water loss could be due to seepage through the streambanks or streambed or possibly a higher rate of evapotranspiration than simulated by FLDAS.*

*P11L18-21 has been updated to read: The unaccounted water loss could be due to seepage under the sand dam wall, through the streambanks, or streambed or possibly a higher rate of evapotranspiration than simulated by FLDAS.*

*P11L25-29 now reads: When evapotranspiration occurs from a sub-surface water table, the rate of evapotranspiration is lower than would be expected if the water table was at the ground surface (Hellwig, 1973). The rate of sub-surface evapotranspiration reduces as the water table retreats farther underground, and the rate of reduction is dependent upon depth and grain size (Hellwig, 1973). Seepage could be contributing to the total loss rate at Chididimo, but there was no evidence of seepage under the dam wall. However, seepage through the streambed could impact the total loss rate at Chididimo.*

**In order to calculate the storage in the sand dam (p.6 line 30-31), why not just assume that it is fully saturated at the end of the wet season? This could be supported if you observe water ponded on top of the trapped sand.**

Thank you for your suggestion. In response to comments from Reviewer 1, we have updated Eq. (1) and initiated the theoretical water loss model at the end of the rainy season. This was updated in the manuscript.

**I am curious as to why the WTMWs were the only attempt to measure water levels the sand dams. Piezometers or even excavated holes in the sand dam could have provided a more comprehensive picture. In my experience observing the water depth in scoop holes that the communities dig can be an excellent indicator of overall water levels, but I don't know if there were present here.**

Borst and de Haas (2006) installed piezometers in the sand dam itself, so we did not think it necessary to repeat this arrangement. Instead, we wanted to focus on how the sand dam affects the water table outside of the stream channel, since the idea that sand dams raise the local groundwater table has really only been explored via modelling (Hut et al., 2008; Quilis et al., 2009). Therefore, we installed WTMWs in the streambanks and the surrounding area to track how the water table was changing over time. We did not consider tracking water levels in the scoop holes dug by community members but will consider this as a viable methodology for future studies.

**I am also not surprised by the results of the macroinvertebrate study. That a dry river bed in an arid region contains no macro invertebrates seems hardly to be a surprise. This methodology seems to be more suited to perennial rivers.**

Boulton et al. (1992), Stubbington et al. (2009), and Verdonschot et al. (2014) all successfully used variations of the methodology described in the manuscript to sample dry river beds. Boulton et al. (1992) even sampled an ephemeral river in the Arizona desert. We believe that we used the appropriate methodology for collecting macroinvertebrates from an ephemeral streambed. If sand dams were less homogeneous and therefore more suitable habitats for macroinvertebrates, we believe we would have been successful in our sampling attempts.

Macroinvertebrate sampling in perennial rivers is most often conducted by placing a net, typically a surber sampler, facing upstream on the streambed and then disturbing the upstream streambed to release any

macroinvertebrates nestled there. The flowing stream then carries the macroinvertebrates into the net, where they are captured and identified.

**There are results on the sediment grain size (p 7 line 19) but no methodology to measure it.**

A formal sediment grain size analysis was not performed. A statement was added to the manuscript to clarify this point.

*p. 7. Line 29-31 has been updated to read: The sand within the sand dam, with the exception of Kimokouwa, was largely a mixture of fine- and coarse-grained sand, as determined by a visual and tactile assessment of the material.*

**p. 8 line 4 – I am struggling to see how the vegetated cover is correlated to the land slope in figure 3. Could this be confirmed through a statistical test?**

The paragraph immediately preceding (p. 7 line 25-p.8 line 2) discusses the relationship between land slope and vegetated cover in the context of Figure 3. This relationship is not immediately apparent from the figure alone. The statement on p. 8 line 4 was edited to reduce the chance for reader confusion, and Pearson's Correlation Coefficient has been calculated and added to the discussion to validate the claim.

*P8L12-19 has been updated to read: That Soweto, the flattest site, and the two transects located in the flattest part of Chididimo display significant increases in vegetation between the dry and rainy seasons suggest that the average percent vegetative cover at a sand dam is correlated to the land slope near the sand dam. The Pearson Correlation Coefficient, ρ, corroborates this observation. The change in vegetative cover between the dry and rainy seasons is negatively correlated (ρ=-0.73) to increasing land slope at the two functioning sand dams, Soweto and Chididimo, indicating that as the land slope increases, the improvement in vegetative cover decreases. The same correlation is not observed at the non-functioning Kimokouwa sand dam (ρ=0.04), which is expected because the sand dam is not contributing to a locally raised water table.*

**p. 8 line 8-9 – point 1 is poorly explained, and again on p. 11 line 29-32. A figure would help here.**

Figure 5, below, was added to the manuscript to help clarify the point. In addition, further discussion of how flat land can generally support more vegetation than steeply sloping land was added to the manuscript. Shemsanga et al. (2018) was added to the manuscript in support of using healthy vegetated cover as an indication of groundwater in Dodoma.

[Figure]

**Figure 5: The roots of plants growing on a (a) steep slope will be farther from the locally raised water table created by a sand dam, and therefore have less access to soil water, than vegetation growing on a (b) gentle slope.**

*P8L24-29 have been updated to read: First, at low elevations above the streambed, groundwater seepage through the streambanks creates a raised water table that is close to the land surface (see Fig. 5). The raised water table has a positive impact on the soil moisture of the unsaturated soil layer, and this additional moisture supports vegetation growth. Second, a lower elevation above the streambed implies a gentler land slope. Gentle slopes give rainwater more time to infiltrate into the soil, because storm surface runoff travels slower over a gentle slope. Increased infiltration results in increased soil moisture and increased recharge of the water table.*

*P8L32-P9L1-7 contain a new paragraph: That the dry season shows a consistent relationship between elevation above the streambed and vegetative cover indicates that the vegetation at Soweto and Chididimo has at least some level of groundwater dependence (see Fig. 4a). The dependence of vegetation on groundwater in arid and semi-arid regions has been well-documented (Elmore et al., 2008; Mata-González et al., 2012; Naumburg et al., 2005; Seeyan et al., 2014; Stromberg et al., 1996; Wang et al., 2011). In arid and semi-arid regions where rainfall is minimal, vegetation often relies on groundwater to supply the additional water needed for plant growth and transpiration (Naumburg et al., 2005). In semi-arid Dodoma, local communities use their knowledge of the relationship between vegetation and groundwater to inform their decisions on where to dig shallow wells (Shemsanga et al., 2018). Therefore, it is reasonable to expect that the vegetative cover at the Soweto and Chididimo sand dams is improved, in part, by the locally raised water table being near the ground surface.*

**p. 9 line 21-25 – this is hard to follow.**

These lines were updated to remove unnecessary detail and clarify the point.

*P10L16-19 have been updated to read: The sand dam at Kimokouwa has a 1.2 m thick silt layer beginning at a depth of 0.5 m that acts as a capillary barrier, inhibiting the infiltration and, therefore, storage of water in the sand dam. As a result, the community is unable to use the sand dam as a source of domestic water. Silt layers formed at the Kimokouwa sand dam, because the dam was improperly constructed for the type of topsoil present in the area (Nissen-Petersen, 2006; de Trincheria et al., 2015).*

**p. 9 line 24 – how can soil be assessed properly in advance to avoid this type of failure?**

A statement with this information was be added to the manuscript.

*P10L19-22 has the following statement added: While the literature on this topic is not well-developed, the soil composition of the streambed before a sand dam is constructed can likely be helpful in determining the distribution of grain sizes a sand dam is expected to capture. This information, coupled with knowledge of the sediment load typically carried by the stream, can inform the need to construct a sand dam's spillway in stages to prevent siltation.*

**p.10 line 31 – by "subsurface water reservoir" do you mean the underlying aquifer of the trapped sand?**

Yes, we mean the water stored between the grains of sand in the sand dam. The wording of this statement has been altered to reduce reader confusion.

*P12L9-10 have been updated to read: However, the dataset does not account for the additional water available in the perched aquifer created by the water stored in the sand dam.*

**p. 12 line 6 – please be more specific on why the stream channel migration is important.**

Noted. We added a few sentences to the manuscript explaining how increased erosion and stream channel migration can lead to the failure of a sand dam.

*P13L18-19 has an additional statement: Severe streambank erosion and/or stream migration can lead directly to sand dam failure by weakening the soil supporting the structure. When this happens, the dam may break or be washed downstream.*

**Additional References**

Anderson, W. B., Zaitchik, B.F., Hain, C.R., Anderson, M.C., Yilmaz, M.T., Mecikalski, J., and Schultz, L.: Towards an integrated soil moisture drought monitor for East Africa. Hydrol. Earth Syst. Sci., 16, 2893–2913, doi:10.5194/hess-16-2893-2012, 2012.

Elmore, A. J., Kaste, J. M., Okin, G. S. and Fantle, M. S.: Groundwater influences on atmospheric dust generation in deserts, Journal of Arid Environments, 72(10), 1753–1765, doi:10.1016/j.jaridenv.2008.05.008, 2008.

Mata-González, R., McLendon, T., Martin, D. W., Trlica, M. J. and Pearce, R. A.: Vegetation as affected by groundwater depth and microtopography in a shallow aquifer area of the Great Basin, Ecohydrology, 5(1), 54–63, doi:10.1002/eco.196, 2012.

Naumburg, E., Mata-gonzalez, R., Hunter, R. G., Mclendon, T. and Martin, D. W.: Phreatophytic Vegetation and Groundwater Fluctuations: A Review of Current Research and Application of Ecosystem Response Modeling with an Emphasis on Great Basin Vegetation, Environmental Management, 35(6), 726–740, doi:10.1007/s00267-004-0194-7, 2005.

Reversing Land Degradation and Desertification: https://www.excellentdevelopment.com/reversing-land-degradation-and-desertification, last access: 1 July 2019.

Sand Dams: https://thewaterproject.org/sand-dams, last access: 1 July 2019.

Seeyan, S., Merkel, B. and Abo, R.: Investigation of the relationship between groundwater level fluctuation and vegetation cover by using NDVI for Shaqlawa Basin, Kurdistan Region – Iraq, Journal of Geography and Geology, 6(3), doi:10.5539/jgg.v6n3p187, 2014.

Shemsanga, C., Muzuka, A.N.N., Martz, L., Komakech, H., and Mcharo, E.: Indigenous knowledge on development and management of shallow dug wells of Dodoma Municipality in Tanzania, Appl. Water Sci. 8, 59, https://doi.org/10.1007/s13201-018-0697-7, 2018.

Stromberg, J. C., Tiller, R. and Richter, B.: Effects of groundwater decline on riparian vegetation of semiarid regions: The San Pedro, Arizona, Ecological Applications, 6(1), 113–131, 1996.

Wang, P., Zhang, Y., Yu, J., Fu, G. and Ao, F.: Vegetation dynamics induced by groundwater fluctuations in the lower Heihe River Basin, northwestern China, Journal of Plant Ecology, 4(1–2), 77–90, doi:10.1093/jpe/rtr002, 2011.

Yilmaz, M. T., Anderson, M.C., Zaitchik, B., Hain, C.R., Crow, W.T., Ozdogan, M., Chun, J.A., and Evans, J.: Comparison of prognostic and diagnostic surface flux modeling approaches over the Nile River basin. Water Resour. Res., 50(1), 386–408, doi:10.1002/2013WR014194, 2014.

---

## Referee Comment (RC3) · Anonymous Referee #3 · 12 Jul 2019

1. An important area of research - I don't think the impact of sand dams on invertebrates has been studied before and there is much that we don't know about such a popular technology in general. The authors are right to ask questions and I would like to see more, more rigourous research.

2. My main feedback relates to the main claim that sand dams are detrimental to macro invertebrates. In order to make this assertion I would have expected to see some sort of comparative study between river beds that are suitable for sand dams but where sand dams have not been built, vs river beds with sand dams.

3. I would not expect to see a great deal of macro invertebrates in a semi arid dry

riverbed, if sand dam levels are not 'normal', what level are they comparing it to?

3. The authors claim a lack of quantitative research skews perceptions towards a rosier picture of sand dams than is true, but their study has a small sample size, over only one season (although intensively studied). There were probably practical limitations - research is expensive, but this could be acknowledged.

———————————————

---

## Author Comment (AC5) · 12 Jul 2019

**Response to Reviewer 3**

We would like to thank the reviewer for the constructive comments on the manuscript. We have considered the reviewer's comments and provide the following responses.

**1. An important area of research - I don't think the impact of sand dams on invertebrates has been studied before and there is much that we don't know about such a popular technology in general. The authors are right to ask questions and I would like to see more, more rigorous research.**

Thank you. We hope this first foray into studying a sand dam's impact on macroinvertebrates leads to further research on the topic and, generally, on macroinvertebrates in semi-arid, ephemeral streams.

**2. My main feedback relates to the main claim that sand dams are detrimental to macro invertebrates. In order to make this assertion I would have expected to see some sort of comparative study between river beds that are suitable for sand dams but where sand dams have not been built, vs. river beds with sand dams.**

The manuscript is careful to not claim that sand dams are detrimental to macroinvertebrates, because the study did not include legitimate control sites due to limitations in implementing the planned research. Rather, the manuscripts uses the phrasing that sand dams "are not suitable habitats" for macroinvertebrates. This was explicitly clarified in the manuscript.

*P7L25-28 has an additional statement: The absence of macroinvertebrates in the sand dams might suggest that sand dams have a negative impact on macroinvertebrate habitat, but it is also likely that sandy streambeds in semi-arid regions are simply inhospitable to macroinvertebrates. Further studies comparing macroinvertebrate assemblages in undammed sandy riverbeds with sand dams are needed to make this distinction.*

**3. I would not expect to see a great deal of macro invertebrates in a semi arid dry riverbed, if sand dam levels are not 'normal', what level are they comparing it to?**

We believe this point is answered by our response to point number 2, above.

**4. The authors claim a lack of quantitative research skews perceptions towards a rosier picture of sand dams than is true, but their study has a small sample size, over only one season (although intensively studied). There were probably practical limitations - research is expensive, but this could be acknowledged.**

Thank you for noting this point. We added a statement acknowledging the limits of our study.

*P2L 29-32 now has an additional statement: The study is limited to only three sand dams, because the study design relies on the active participation of local community water groups. Most of the community water groups formed during sand dam construction had disbanded at the time of this study. The breadth of the study was further limited by long travel times between sites and difficulties related to equipment access.*

---

## Author Response (AR2)

**Response to Reviewer 1**

We would like to thank dr. ir. Rolf Hut for his constructive comments on the manuscript. We have considered his comments and provide the following responses.

**Regarding the water balance model, I remain unconvinced that…the conclusion regarding ET can be drawn the way you do.**

The uncertainty in the model remains, but we believe the updates to the model and calculations made at the request of Reviewer 3 will satisfy some of your misgivings regarding our conclusions. Changes in the calculations for the water balance model are described in the response to Reviewer 3.

We appreciate your continued scientific skepticism, as it has allowed us to refine our model and sharpen our discussion throughout this review process.

**Response to Reviewer 2**

We would like to thank Dr. Alison Parker for her constructive comments on the manuscript. We have considered her comments and provide the following responses.

**I think it would be useful if they mentioned in the text that only three of the Tanzanian sand dams still have community water groups.**

*P3L12-13 has been updated to read: Only three of the community water groups formed during sand dam construction remained active at the time of this study.*

**They are relying on Hellwig (1973) to calculate evaporation from bare sand. (…) I would urge the authors to consider the new interpretations presented in: Quinn, R, Rushton, K., Parker, A, (2018) Evaporation from bare soil: lysimeter experiments in sand dams interpreted by numerical models, Journal of Hydrology 464, 909-915.**

We did not calculate evaporation from bare sand using the methods suggested by Hellwig (1973). Estimates for ET were taken from the FLDAS dataset. Use of FLDAS has been clarified in the text (also at the request of Reviewer 3). Quinn et al. (2018b) has been added to the manuscript references, and the content was used to inform the discussion of ET losses, particularly for the three stages of ET loss rate. Thank you for bringing this manuscript to our attention.

*P13L21-24 have been updated to read: The Soweto dam exhibits three distinct phases of water loss: shallow ET, deep ET, and minimal ET (see Fig. 9b; Quinn et al., 2018b). The minimal ET phase occurs during the period in which the community water group indicated they were no longer able to abstract water from the sand dam. At this point, the water table has retreated too far underground for the community to draw water and, at this depth, the rate of ET is likely negligible (Hellwig, 1973; Quinn et al., 2018b).*

*P14L1-2 have been updated to read: However, the dataset does not account for the depth at which the sand dam water is stored or for the unique features, such as wind speed, topography, vegetation, and shading, that impact ET rates (Hellwig, 1973; Quinn et al., 2018b).*

**Response to Reviewer 3**

We would like to thank Dr. Tibor Stigter for his constructive comments on the manuscript. We have considered his comments and provide the following responses.

**My main remark is that the collected groundwater level data allow a more comprehensive overview of the water table response at the three sand dam sites, including the non-functioning one. A total of 63 wells were drilled and water levels were monitored over one year (most successfully at Chididimo), which gives much more insight into the water table dynamics (in time and space) than described and discussed in the text. Moreover, the borehole logs from the 63 wells provide valuable lithological information for understanding connectivity between the (sand) river and its banks, which is now poorly addressed. The other component that I find a bit missing is the water quality aspect, which links directly to the apparently dominant mechanism of water loss in the areas, namely evapotranspiration (ET). If ET is indeed that dominant, as compared to lateral or vertical seepage, this should be seen in the increasing salinity of the water throughout the dry season, especially after a few years (the Soweto sand dam for instance was completed in 2011). It would be interesting to see if there is any additional information from EC measurements or interviews with the community that evidence a problem of increasing salinity.**

A future work section has been added. After going through your comments, we realized that our 4ᵗʰ research question in the introduction and word choice throughout the manuscript were misleading. We frequently mentioned "water table" and "water table dynamics," but our analysis was focused on partitioning out the various loss mechanisms and discussing the factors impacting them. We have altered the language of our manuscript in many places to more accurately represent our analysis.

*P16L1-7 have been added: Future analysis of the collected dataset will focus on exploring the spatial variability in the local geology and its interactions with groundwater in the vicinity of the sand dam. Water table dynamics will be investigated in conjunction with the variability of evapotranspiration in and around the sand dams. In addition, the change in vegetative cover relative to groundwater depth will be studied using both the field measurements detailed here and Normalized Difference Vegetation Index. Future sand dam research should also investigate water quality in the sand dams in the context of increasing salinity as evidence for or against high rates of evapotranspiration.*

**Abstract**

**Is well-written, but will need some editing after including additional results and discussion. It would be useful to read here why sand dams are not a suitable habitat for macroinvertebrates.**

*P1L9-10 have been updated: The study investigated a sand dam's impact on macroinvertebrate habitat, vegetation, and streambank erosion and explored a sand dam's water loss mechanisms.*

*P1L13-14 have been updated: The study found that sand dams are too homogeneous to provide the sustenance and refugia macroinvertebrates need at different life stages*

**Introduction**

**The response of the groundwater table in the surrounding area of a sand dam largely depends on the connectivity between that river margins and the sand river. This connectivity is often seen to be limited, which in part explains why the effect of an increasing water table is very local. This is not addressed in the text. The other aspect I would like to see introduced is that of sand dam water quality and processes (and/or human activities) that affect it.**

A brief paragraph summarizing these relationships has been added to the introduction section.

*P2L8-17 have been added: Sand dams are small, reinforced concrete dams built atop impermeable streambeds in arid regions with infrequent, high-intensity rainfall (see Fig. 1). The high-intensity rainfall*

*erodes soil from the land surface and deposits the coarser particles, usually sand, upstream of the dam. The sand stores primarily flash flood-water, where it is naturally filtered, protected from evaporation, and helps raise the groundwater level in the surrounding area due to recharge from the increased subsurface storage (Borst and de Haas, 2006; Hut et al., 2008; Quilis et al., 2009). The extent of a sand dam's impact on the groundwater level, however, is limited by the geologic connectivity between the sand dam and the riparian zone and by the community's water use rate (Hut et al., 2008; Quinn et al., 2019). While a sand dam does filter water in a process similar to a slow sand filter, water abstracted from sand dams via scoopholes and covered wells exceeds World Health Organization recommendations for turbidity (73 % exceedance), conductivity (24 % exceedance), and thermotolerant coliform concentration (55 % exceedance) (Quinn et al., 2018a).*

**Study area**

**In the description of climate I would like to see average annual values of P, T and PET.**

A few sentences with the requested information has been added to the study area.

*P3L25-28 have been added: The average annual rainfall for Dodoma is 601 mm, and the potential evapotranspiration is 1800 mm. The average annual temperature in Dodoma is 23.0° C. The average annual rainfall for Longido is 696 mm, and the potential evapotranspiration is 1640 mm. The average annual temperature in Longido is 20.7° C (Platts et al., 2015).*

**The description of the geology is lacking. This is crucial information to discuss the connectivity between the (sand) river and its banks. As soil logs were collected from each of the 63 drillings, there is a lot of information available.**

A brief description of the site geology for each sand dam was added to the study area.

*P4L9-11 have been added: The soil deposited behind the Kimokouwa sand dam is largely silty sand with thick silt layers interspersed. In the riparian zone, the soil is primarily reddish sandy clay.*

*P4L17-18 have been added: The soil deposited behind the Soweto sand dam is moderately sorted sand, and the riparian zone is predominantly silty sand.*

*P4L25-26 have been added: The soil deposited behind the Chididimo sand dam is moderately sorted sand. The riparian zone contains primarily silty sandy gravel.*

**The characterization of the dimensions of the sand reservoirs built up behind the sand dams is missing. It would be important to state something about thickness, length and estimated storage capacity. In combination with the extinction depth this reveals the depth of "safe storage" not affected by ET. Only the maximum width (at the dam site) has been indicated for each area.**

Unfortunately, MCC did not have any plan documents from when the sand dams were constructed, and we did not probe the bottom of the sand dams to generate cross sections. Therefore, we do not know the stream slope pre-dam. Based on the scoopholes dug by community members at Soweto and Chididimo and the depth achieved in the installed WTMWs, we estimate the depth of the sand dams to be around 2-3 m. At Kimokouwa, an exploratory borehole drilled just upstream of the sand dam reached a maximum depth of 2.6 m. These estimates for sand dam depth are consistent with those measured by Quinn et al. (2019) and are less than the 4-6 m estimate provided in Aerts et al. (2007).

*Table 1 has been updated to show the requested information.*

**Table 1:** Physical parameters of the three sand dams

| Sand Dam | Total width (m) | Total length (m) | Spillway (m) | Estimated storage volume* (m$^3$) | Wing walls (m) | Spillway height (m) |
|---|---|---|---|---|---|---|
| Kimokouwa | 28.78 | 150 | 8.74 | 1310 | 20.04 | 2.06 |
| Soweto | 23.96 | 350 | 16.95 | 5930 | 7.01 | 1.27 |
| Chididimo | 22.71 | 300 | 9.60 | 2880 | 13.11 | 1.30 |

*Note: Storage volume estimated using an average sand dam depth of 2.5 m and porosity of 0.40. The spillway is approximately equal to the width of the stream channel.

*P4L3-4 has been updated to read: All three sand dams vary in their construction specifications, length, and storage capacity (see Table 1).*

**An overall schematic diagram of a sand dam, and how it matures over time, would also be very useful. This would further help explain the non-functionality of the Kimokouwa sand dam.**

*The requested figure has been added as Figure 1.*

**The location of the hand pumps is not clear for the Kimokouwa and Soweto sand dams (where is "near the sand dam"?). For Chididimo why was the hand pump installed so far away from the dam?**

Statements with this information have been added to the study area descriptions.

*P4L10-11 have been updated to read: A hand pump was installed in the right bank, 30 m upstream of the sand dam in April 2016.*

*P4L18-19 have been updated to read: A hand pump was installed in the left bank, 85 m upstream of the sand dam at the time of dam construction.*

*P4L26-28 have been updated and expanded to read: A hand pump was installed within the stream channel 150 m upstream of the sand dam at the time of dam construction. The community selected this site for the hand pump, because they were able to extract water from the sandy streambed at that location before the sand dam was constructed.*

**Why was there no observation well placed directly behind (upstream of) the sand dam, in the middle of the river? This would have allowed to measure the fluctuations that are not affected by transpiration (only evaporation, seepage and abstraction).**

Fair point, and perhaps a WTMW directly behind the sand dam would have yielded interesting results. However, we chose not to install any WTMWs directly in the sand dam. This has been done for other studies (see Borst and de Haas, 2006; Quinn et al., 2019), but we did not want to disrupt the functions of the sand dam or cause any potentially negative side effects with our research.

**For the water table monitoring wells (WTMW) was topographic elevation of the top of the pipe measured, and with what accuracy?**

A statement with this information has been added to section 3.5 Water table monitoring.

*P6L31-32 have been added: The elevation of the top of the WTMW pipes was measured relative to the ground surface with a tape measure, accurate to the nearest cm. The ground elevation at the WTMWs was determined with a calibrated GARMIN GPSMAP 64s.*

**The water balance and resulting volumetric water loss calculations require some clarification. It is not clear how the "control volume" and corresponding water volume were calculated. I understand from the text that water volumes were integrated over the area covered by the monitoring wells. But how did the authors take the spatial variation in texture and related porosity into account?**

This was not clearly explained in the manuscript. A paragraph describing the calculations has been added. The calculations were adjusted to account for variations in soil texture, altering our weekly water volumes figure (now Fig. 9) and resulting conclusions.

*P7L17-25 have been added: The weekly average height of subsurface water in each WTMW was calculated from the field data, accounting for the difference in soil porosity between the sand dam and the riparian zone. A value of 0.42 was used for the porosity in the sand dam; 0.40 was used for the porosity in the riparian zone (Rawls et al., 1982). Inverse distance weighting interpolation was applied to create uniformly spaced grids of average water height at a weekly time step. The weekly average water volume was then calculated by multiplying the water height grids by the grid spacing and summing across the control area. The control area is the portion of the study area enclosed by the installed WTMWs (see Fig. 2 c,d). For Chididimo, this area is 32 274 $m^2$, while it is 41 995 $m^2$ for Soweto. The weekly average control area water volume calculated from the field data is compared to a theoretical weekly average water volume, described below.*

**How did the texture of the margins compare with that of the river bed?**

We believe this question was suitably addressed in our response to your inquiries about site geology. The changes made are included again for ease of reference.

*P4L9-11 have been added: The soil deposited behind the Kimokouwa sand dam is largely silty sand with thick silt layers interspersed. In the riparian zone, the soil is primarily reddish sandy clay.*

*P4L17-18 have been added: The soil deposited behind the Soweto sand dam is moderately sorted sand, and the riparian zone is predominantly silty sand.*

*P4L25-26 have been added: The soil deposited behind the Chididimo sand dam is moderately sorted sand. The riparian zone contains primarily silty sandy gravel.*

**What was the spatial distribution of the water table and the spatial variation in water table response to the sand river. Did the water table in the margins actually respond to the river level recharge or to aerial recharge from rainfall? This could be shown through piezometric maps at different moments of the year (end of the wet season and dry season) to check the losing or gaining behaviour of the sand dam. With so many well level recordings in time and space this invaluable information could be provided. It will also allow showing the importance of ET vs. direct soil evaporation (at places and/or times in the year where/when vegetation is absent). It will further allow to assess if lateral flow towards the river margin occurs, as hypothesized by the authors.**

Thank you for your comment and ideas for additional analysis. We agree that this is valuable information and an interesting opportunity to expand the findings from the data we collected. We are conducting a more thorough analysis of the water table/sand dam dynamics for another manuscript, however, so these findings will not be included in this manuscript. We chose to limit the water table discussion here to keep the manuscript to a reasonable length, because we also covered macroinvertebrates, vegetation, and erosion at the sand dams. As written, the manuscript is intended to provide information on the data collected and discuss the overall trends in that data. We realize that the fourth research question listed in the Introduction section is misleading, so it has been updated to reflect the water storage and loss analysis presented in this manuscript.

*P3L4-5 have been updated to read: (4) What are the dominant mechanisms driving water loss from a sand dam and the riparian zone?*

**FLDAS calculations should be better explained. In section 4 the authors mention that the FLDAS dataset calculates evaporation from bare soil based on simulated soil moisture content. How does this compare to transpiration, and how is the reduction of E and T with depth taken into account?**

Additional information on the FLDAS dataset was added to the manuscript, with particular focus on how the dataset calculates transpiration and evaporation.

*P7L28-31 have been added: FLDAS is a set of models designed to provide accurate climate estimates for the purpose of drought monitoring in data-sparse regions susceptible to food and water security issues (McNally et al., 2017). FLDAS provides daily and monthly climate data consisting of 25 different variables for Western, Eastern, and Southern Africa.*

*P8L18-22 have been added: FLDAS calculates transpiration by scaling potential evapotranspiration in proportion to solar radiation, vapor pressure deficit, air temperature, and soil moisture. Evaporation from bare soil in the FLDAS dataset is calculated by scaling potential evapotranspiration based on current soil moisture (McNally et al., 2017). Therefore, the rates of transpiration and evaporation in the FLDAS dataset will decrease as the water table retreats from the ground surface and soil moisture declines*

**Canopy-interception of rainfall should be entirely excluded from the water loss calculations, unless you include rainfall as an input to the water balance. If only ET from the subsurface is considered (water table, soil moisture) then including canopy interception makes no sense, even in the riparian corridors.**

This is correct. The description of Eq. (1) has been updated to reflect this, and the calculations have been re-done to apply a modifier of 0.85 to the FLDAS ET data for the entire control volume.

*P8L2-5 has been updated to read: where $Q_{out,dry\ season}$ is the rate of water loss from the sand dam after the end of the rainy season, E is total evapotranspiration modified by α, which is 0.85, $Q_{sb}$ is baseflow-groundwater runoff, and $Q_{com}$ is the community's water use.*

*P8L14-15 has been updated to read: Eq. (1) is only applied during the dry season, and therefore the control volume will not lose water due to evaporation of canopy-intercepted rainfall.*

*P8L18 has been updated to read: Therefore, total evapotranspiration is reduced by 15 % in Eq. (1), resulting in an α of 0.85.*

**Please explain how Qsb(t) (baseflow-groundwater runoff) was determined in the field. It is not mentioned.**

Qsb(t) was not determined in the field. The value used for Qsb was taken from FLDAS data. A sentence has been added to clarify this.

*P8L3-5 have been added: E and $Q_{sb}$ are taken directly from the FLDAS dataset (McNally et al., 2017), while $Q_{com}$ was calculated based on each community's accounting of their water use.*

**Sand dam impact**

**I recommend to name this section Results and Discussion, which you then separate per topic. Section 5 could then be General Discussion and Considerations, and then the Conclusions section is perhaps not really necessary.**

The sections were re-named, as suggested. The conclusion section has been removed in favor of a Future Work section.

*P8L28 now reads: 4 Results and Discussion*

*P14L22 now reads: 5 General Discussion and Considerations*

*P16L1 now reads: 6 Future Work*

**4.2 Vegetation: The authors provide evidence to support the statement that "it is reasonable to expect that the vegetative cover at the Soweto and Chididimo sand dams is improved, in part, by a locally raised water table near the ground surface". The arguments are clear, but could be further supported by the water level measurements, which are not shown.**

We thank the reviewer for his ideas on how to strengthen our arguments. However, we are struggling to include this information on our existing figures without compromising their clarity. We will consider showing the relationship between depth to the water table and vegetated cover in future manuscripts.

**I do wonder why at the lowest elevation vegetation is most sparse in the dry season.**

We believe this is because the lowest elevation vegetation shown in Fig. 5a is below the ordinary high water mark for the streams. Vegetative growth below the ordinary high water mark is inhibited, but the abundance of water available at the sand dam margins during the wet season gives rise to rapid plant growth. This relationship is mentioned in P10L12-13: "As Fig. 5a indicates, there is low vegetative cover right at the stream edge (lowest elevation), which signifies streamflow frequently rising above this point and inhibiting vegetation growth."

**Moreover, I do not agree that the fact that Kimokouwa has a very low vegetation cover in both the dry and wet season is related to a poorly functioning sand dam. It seems much more related to the topography, with higher elevations and slopes, and hence higher runoff and lower infiltration potential. For vegetation the role of groundwater can be important, but the role of the infiltration and water-holding capacity of the soil can be at least as relevant. In finer soils and flatter topography you will therefore expect more vegetation. The role of the sand river is questionable, with a few exceptions shown in Figure 3 (VT4 wet season Soweto, VT2 dry season Chididimo, and perhaps VT3-4 wet season Kimokouwa).**

You are correct, land slope and soil play a big role in land cover. A statement acknowledging such was added.

*P10L31-32 has been added: This may be due solely to the impact of the sand dams, but it is equally likely that the steeper slopes and finer soils at Kimokouwa impact its vegetative cover.*

*P10L34 has been added: Land slope and soil, however, must also be considered*

**4.3 Streambank erosion: clear and well supported by the observations; I am only not sure why the downstream curve for Chididimo is missing.**

As mentioned in section 3.4 Erosion Study, erosion pins were only placed at one location at Chididimo. This sentence has been updated for clarity.

*P6L9-11 were updated to read: Pins were installed at fewer locations at Chididimo, because the stream did not have a clearly defined bank, and, where present, the streambank was often too rocky to permit insertion of the pins.*

**4.4 Water Table: As mentioned, this section could be enriched by showing the results of the 12-hourly measurements taken at Chididimo and Soweto, to show the temporal and spatial variations of the**

**water table and how they related to the sand dam water level. The borehole log data could be shown to address the connectivity between river and banks, and the potential for groundwater flow into the banks. In many cases this connectivity is actually rather limited, even when sand dam water level is able to rise above the regional water level in the margins. That is probably another reason why the sand dam at Kimokouwa was non-functional, as the river valley is steeply incised (as in figure 5a), creating a small narrow sand reservoir with limited storage capacity and no possibilities to actually feed the neighbouring water table, which is expected to be above the sand dam level. Again, this could be verified with the water level data where they are available.**

We appreciate your ideas for further analysis. We believe this content is more appropriate for a second paper, as mentioned previously. You are correct that limited connectivity between the Kimokouwa sand dam and its riparian zone could explain why the dam is non-functioning. This is supported by the fact that none of the Kimokouwa WTMWs had water, except for the one WTMW closet to the dam wall.

*P12L3-6 have been added: Kimokouwa sand dam's water storage is also likely limited by the poor connectivity between the silty sand in the channel and the reddish clay that dominates the riparian zone. Groundwater is unable to travel freely between the sand dam and the riparian zone, as evidenced by the absence of water in all but one WTMW.*

**How is the 1% of the total water lost attributed to baseflow determined?**

The volume of total water lost is based on the field data calculations. The portion of the water lost attributed to baseflow is the $Q_{sb}(t)$ term in Eq. (1), which comes directly from the FLDAS dataset. We have added a sentence to section 3.5 to clarify that the $Q_{sb}(t)$ term comes from FLDAS.

*P8L3-6 have been added: E and $Q_{sb}$ are taken directly from the FLDAS dataset (McNally et al., 2017), while $Q_{com}$ was calculated based on each community's accounting of their water use.*

**In Figure 8 show how the red line was calculated and add the ET line for comparison.**

The caption of (now) Figure 9 includes a description of how to calculate the theoretical line. We added a bit more detail to the explanation and believe this is adequate to inform the reader of the basis for the theoretical line.

*Figure 9 caption now reads: The theoretical line, initiated at the end of the rainy season, shows the theoretical volume of water in the study area, calculated by integrating Eq. (1) and subtracting from the field-determined volume of water at the end of the rainy season.*

The theoretical ET line was added to (now) Figure 9. A description of the line was added to the caption.

*Figure 9 caption now includes: The theoretical ET line shows the portion of total theoretical loss attributed to ET in the FLDAS dataset.*

**How does ET vary spatially in the areas? This is shown by the water table fluctuations? How do the variations below vegetative cover (dominated by ET) compare to those below bare soil (in particular the sand reservoir area, dominated by E)?**

There is likely not much variety in ET across the Chididimo study area, because Chididimo is uniformly vegetated with natural vegetation and in a uniform stream valley. Conversely, the Soweto study area has large plots under cultivation, some sparsely vegetated areas, and some areas with dense natural vegetation and is topographically varied.

We are not convinced that our collected water table data can be used to accurately calculate ET in the study area, hence the adoption of FLDAS ET data for this analysis. Community volunteers recorded WTMW measurements to the nearest cm. Therefore, the data lacks the precision of typical ET data. In addition, because the sand dam represents an open system with multiple sources of loss, we cannot necessarily attribute all water depth changes measured at the wells to ET. Community use, baseflow, and seepage through cracks in the bedrock or under the sand dam all contribute to the declining water table. To determine how ET varies spatially across the study areas, we should have installed a series of lysimeters.

We will investigate whether conclusions can be drawn about the spatial variability of ET when we conduct a more thorough analysis of water table dynamics around the sand dam for our planned second manuscript. However, as mentioned above, the uncertainty of these calculations is likely to be an issue.

**The fact that Chididimo and Soweto experience approximately 100 mm lower rainfall than in Kenya is not the cause of faster storage depletion throughout the dry season. It is the length of the rain season and the duration/frequency of river flow that determines whether a sand dam lasts longer. This is actually addressed in the discussion.**

That is true. The text in section 4.4 has been updated to clarify why the lower annual rainfall in Dodoma limits the sand dams' potential as a water resource.

*P14L175-20 have been updated: At Chididimo and Soweto, this is simply not the case. Chididimo and Soweto experience approximately 100 mm lower annual rainfall in one, four-month rainy season and higher rates of ET than a typical sand dam in Kenya experiences during two rainy seasons (NASA/GSFC/HSL, 2016). Therefore, the Dodoma sand dams have lower potential for storing water than their Kenyan counterparts.*

**The authors state correctly that "The Soweto sand dam is losing water during the shallow ET phase at nearly 2.5 times the rate of the Chididimo sand dam." They argue that this is because: "The width of the Soweto sand dam is nearly twice that of Chididimo, providing a greater surface area of sand from which evaporation occurs". However, the rate is already given in mm/day, independent of area. It is more likely, also by looking at the elevations in Figures 1c/1d, that the water levels are overall much shallower in the Soweto area. This would also show in the water table monitoring data. In that case, the grey area in Figure 8a would actually correspond to "Deep ET" or "Intermediate ET" in Chididimo, rather than shallow ET. The difficulty here is that for the calculations not only the sand dam itself is considered, but also the surrounding area. It would be interesting to compare this to the calculations only for the sand reservoir behind the sand dam.**

We thank the reviewer for his comment and acknowledge that, as written, the text could be confusing. We rearranged the paragraph, as detailed below, and removed the reference to seepage. As an additional note, the water levels at Soweto are, on average, approximately 0.5 m deeper than at Chididimo. Chididimo, indeed, has shallower groundwater levels near the sand dam compared to Soweto.

*P14L7-14 were rearranged: The Soweto sand dam is losing water during the shallow ET phase at more than twice the rate of the Chididimo sand dam. The combination of stream width and vegetative cover contribute to Soweto's higher rate of water loss. The width of the Soweto sand dam is nearly twice that of Chididimo, providing a greater surface area of sand from which evaporation occurs (see Table 1). For an equivalent water content, sub-surface evaporation rates from sand in the sand dam are higher than from the loamy soils in the riparian zone due to the differences in soil suction (Wilson et al., 1997). Also, different types of vegetation transpire water at different rates (Lautz, 2008). The banks of Chididimo are generally covered with natural vegetation, whereas the Soweto community intensively cultivates the banks. Natural*

*vegetation in a semi-arid climate requires less water than cultivated crops, therefore the rate at which the Soweto vegetation transpires water contributes to Soweto's rapid water loss.*

**5. Discussion**

**The authors mention that "the functioning sand dams had significant impact on the local water table", but do not provide the evidence to support this. This would require knowledge on the water table dynamics before the sand dams were constructed, or the thickness of the sand deposits/total capacity of the sand reservoir before the sand dam was built. It is obvious that there will have been impact, but at least try to show it through the observed water level behaviour after the dam was built, showing the extent to which the sand dam had an effect.**

You are correct, the evidence provided in this manuscript does not support this statement. Because we are not intending to cover the water table dynamics in this manuscript, we have adjusted this statement to reflect the presented results.

[revised manuscript text omitted]